# A multilayer approach and its application to model a local gravimetric quasi-geoid model over the North Sea: QGNSea V1.0

**Yihao Wu**[1,2], **Zhicai Luo**[3], **Bo Zhong**[4], **Chuang Xu**[2,5]

[1] School of Earth Sciences and Engineering, Hohai University, Nanjing, China

[2] State Key Laboratory of Geodesy and Earth's Dynamics, Institute of Geodesy and Geophysics, Chinese Academy of Sciences, Wuhan, China

[3] MOE Key Laboratory of Fundamental Physical Quantities Measurement, School of Physics, Huazhong University of Science and Technology, Wuhan, China

[4] School of Geodesy and Geomatics, Wuhan University, Wuhan, China

[5] School of Civil and Transportation Engineering, Guangdong University of Technology, Guangdong, China

*Correspondence to*: Yihao Wu (yihaowu@hust.edu.cn) and Chuang Xu (chuangxu@hust.edu.cn)

**Abstract:** A multilayer approach is set up for local gravity field recovery within the framework of multi-resolution representation, where the gravity field is parameterized as the superposition of multiple layers of Poisson wavelets located at different depths beneath the Earth's surface. The layers are designed to recover gravity signals at different scales, where the shallow and deep layers mainly capture the short- and long-wavelength signals, respectively. The depths of these layers are linked to the locations of different anomaly sources beneath the Earth's surface, which are estimated by wavelet decomposition and power spectrum analysis. For testing the performance of this approach, a gravimetric quasi-geoid model over the North Sea, QGNSea V1.0, is modeled and validated against independent control data. The results show that the multilayer approach fits the gravity data better than the traditional single-layer approach, particularly in regions with topographical variation. An Akaike information criterion (AIC) test shows that the multilayer model obtains a smaller AIC value and achieves a better balance between the goodness of fit of data and the simplicity of the model. Further, an evaluation using independent GPS/leveling data tests the ability of regional models computed from different approaches towards realistic extrapolation, which shows that the accuracies of the QGNSea V1.0 derived from the multilayer approach are better by 0.4 cm, 0.9 cm, and 1.1 cm in the Netherlands, Belgium, and parts of Germany, respectively, than that using the single-layer approach. Further validation with existing models shows that QGNSea V1.0 is superior with respect to performance and may be beneficial for studying ocean circulation between the North Sea and its neighboring waters.

# 1. Introduction

Knowledge of the earth's gravity field at the regional scale is crucial for a variety of applications in geodesy. It not only facilitates the use of the Global Satellite Navigation System to determine orthometric/normal heights in geodesy and surveying engineering, but also plays a fundamental role in oceanography and geophysics.

Regional gravity field determination is typically conducted within the framework of the remove-compute-restore methodology (RCR) (Sjöberg, 2005), where long-wavelength signals are often recovered by satellite-only global geopotential models (GGMs) derived from dedicated satellite gravity missions such as the GRACE (Gravity Field and Climate Experiment) (Tapley et al., 2004) and GOCE (Gravity Field and Steady-State Ocean Circulation Explorer) (Rummel et al., 2002). Middle- and short-wavelength signals are extracted from locally distributed gravity-related measurements (Guo et al., 2010; Wang et al., 2012; Wang et al., 2018). Spherical radial basis functions (SRBFs) have become of great interest for gravity field modeling at the regional scale in the recent years (Eicker et al., 2013; Naeimi et al., 2015). Typically, the most commonly SRBFs are implemented using the single-layer approach, i.e. the parameterization of the gravity field is based on only a single-layer of the SRBF grid (Wittwer, 2009; Bentel et al., 2013; Slobbe, 2013; Wu et al., 2016a).

It has been suspected for long that the single-layer approach may fail to extract the full information contains in local gravity data, thus the multi-resolution representation (MRR) method with SRBFs has been investigated in the recent years (Freeden et al., 1998; Fengler et al., 2004, 2007). Freeden and Schreiner (2006) proposed a multi-scale approach based on locally supported wavelets for determining regional geoid undulations from deflections of the vertical. Further, Freeden et al. (2009) demonstrated that a multi-scale approach using spherical wavelets provided a powerful technique for the investigation of, local fine-structured features such as those caused by plumes, which allowed the scale- and space-dependent characterization of this geophysical phenomenon. Schmidt et al. (2005, 2006, 2007) developed a multi-representational method for static and spatiotemporal gravitational field modeling using SRBFs, where the input gravity signals were decomposed into a number of frequency-dependent detail signals; they concluded that this approach could improve the spanning fixed time intervals with respect to the usual time-variable gravity fields. Chambodut et al. (2005) set up a multi-scale method for magnetic and gravity field recovery using Poisson wavelets and created a set of hierarchical meshes associated with the wavelets at different scales, where a level of subdivision

corresponded to a given wavelet scale. Panet et al. (2011) extended the approach developed by Chamboudt et al. (2005) by applying a domain decomposition approach to defining the hierarchical subdomains of wavelets at different scales; this enabled the splitting of a large problem into smaller ones. The results of these studies show that the multi-scale approach using SRBFs has a good potential for gravity field recovery. However, to the best of our knowledge, no direct comparisons have been made between the single-layer approach and the multi-scale one regarding their performances in local gravity field recovery. Further, the existing multi-scale methods mainly construct the multi-scale framework in a mathematical sense, and no explicit geophysical meanings are investigated. In this study, inspired by the power spectral analysis of local gravity signals, we develop new parameterizations of the SRBF network within the MRR approach. In this approach, multiple layers are linked to the anomaly sources at different depths beneath the Earth's surface, and the aim is to recover the signals with different spectral contents. Moreover, the performances of the multilayer approach and traditional single-layer approach are directly compared, and the advantages and disadvantages of the two methods are analyzed.

The structure of the manuscript is as follows. The study area and data collection methods are described in Section 2. Then, the MRR method with SRBFs is introduced, where Poisson wavelets that have band-limited properties are chosen as the basis functions. Wavelet decomposition and power spectrum analysis are applied in constructing the network of Poisson wavelets. In addition, the function model based on this multilayer approach is derived and the method for estimating the unknown coefficients of Poisson wavelets is introduced. The construction of the multilayer model is described in Section 3. The performances of the two approaches (single-layer and multilayer) are also compared in this section. Finally, a gravimetric quasi-geoid over the North Sea, called QGNSea V1.0, is modeled using the multilayer approach and compared with other models for cross validation. We present the summary and the main conclusions of this study in Section 4.

## 2. Data and methods

### 2.1. Study area and data

A region in Europe, from 49 °N to 61 °N and -6 °E to 10 °E, covering the mainland of the Netherlands, Belgium, parts of the North Sea, UK, Germany, and France is chosen as a case study. Data regarding point-wise terrestrial and shipborne gravity anomalies are used in this study, which were provided by different institutions. The details of the data pre-processing procedures can be found in Wu et al. (2017c), where crossover adjustment and low-pass filters were

applied to remove systematic errors and reduce high-frequency noise, respectively. Since the terrestrial and shipborne gravity data were derived from different institutions over various time spans, the horizontal and vertical datums need to be unified. The European Terrestrial Reference System 1989 (ETRS89) and European Vertical Reference Frame 2007 (EVRF2007) are chosen as the horizontal and vertical systems, respectively (Slobbe, 2013). Datum transformations are performed on all of the data following the methods proposed by Wu et al. (2017c). Moreover, the long-wavelength signal content in the data is reduced by removing the contribution of the GOCO05s global geopotential model complete to degree and order (d/o) of 280 (Mayer-Gürr et al., 2015). At the very short wavelengths, residual terrain model (RTM) is applied. The details of the RTM reduction process and the residual gravity data can be found in Wu et al. (2017c).

## 2.2. Multilayer approach

According to Schmidt et al. (2006, 2007), the MRR of the Earth's disturbing potential $T(z)$ at position $z$ is expressed as

$$T(z) = \bar{T}(z) + \sum_{i=1}^{I} t_i(z) + \delta(z) \tag{1}$$

where $\bar{T}(z)$ represents a reference model, e.g. a global geopotential model (GGM) computed from spherical harmonics; $\delta(z)$ represents unmodeled signals; $I$ is the number of levels (resolutions); $t_i(z)$ is the detailed signal of level $i$, and the higher the level value $i$ is, the finer are the structures extractable from the input data; $t_i(z)$ is computed as a linear combination of SRBFs (Schmidt et al., 2007).

$$t_i(z) = \sum_{n=1}^{N_i} \beta_{i,n} \Psi_i(z, y_{i,n}) \tag{2}$$

where $\Psi(z, y)$ is the SRBF, $N_i$ and $\beta_{i,k}$ are the number and unknown coefficient of the SRBF at level $i$, respectively, and $y_{i,n}$ is the position of the SRBF at this level.

The reference GGM and RTM corrections are removed from the original data to decrease the signal correlation length

and smooth the data (Omang and Forsberg, 2000). Then, only the residual disturbing potential $T_{res}(z)$ is parameterized by the SRBF using the MRR approach. Ignoring the unmodeled signals, the residual disturbing potential is expressed as a series of detailed signals at different levels, combining eq. (1) and eq. (2)

$$T_{res}(z) = \sum_{i=1}^{I} \sum_{n=1}^{N_i} \beta_{i,n} \Psi_i (z, y_{i,n}) \tag{3}$$

where $\Psi_i$ is computed as the difference between the spherical scaling functions with low-pass filter characteristics corresponding to consecutive levels $i+1$ and $i$; $\Psi_i$ can also be expressed as the SRBF with band-limited properties in the frequency domain (Schmidt et al., 2007). $\Psi$ represents Poisson wavelets with band-limited properties (Chambodut et al., 2005) in this study, the complete definition of which can be found in Holschneider and Iglewska-Nowak (2007).

Poisson wavelets can also be identified as the multipoles inside the Earth, and the scales of Poisson wavelets can be linked to their depths beneath the Earth's surface. These depths are the key parameters in determining wavelet properties in space and frequency domains (Chambodut et al., 2005). The detailed signal at level $i$ in eq. (2) can be estimated using a linear combination of Poisson wavelets located at a specific depth. Poisson wavelets at various depths demonstrate different properties in the frequency domain. At shallow depths, the scales decrease, and wavelet spectrums shift toward the high degrees of the spherical harmonics (SH) and become more sensitive to local signal features with high-frequency properties, and vice versa (Chambodut et al., 2005). Moreover, Poisson wavelets at different depths can be linked to the detailed signals at various levels, and are sensitive to the various spectral contents of input signals. They can be used for multi-resolution representation. These properties are crucial for local gravity field modeling. The residual disturbing potential is typically a band-limited signal within the RCR framework, and Poisson wavelets with band-pass filter characteristics are preferable for band-limited signal recovery (Bentel et al., 2013).

Rather than as an MRR, we interpret eq. (3) as the multilayer approach that takes into consideration that Poisson wavelets at different depths have different characteristics, and different layers correspond to Poisson wavelets' grids at various depths. We place the Poisson wavelets in the Fibonacci grids under the Earth's surface, and keep these grids

parallel with the Earth's surface (Tenzer et al., 2012). Instead of associating the Poisson wavelets at different depths to the hierarchical meshes with various levels (Chambodut et al., 2005), we apply a wavelet analysis approach to estimate the depths of multiple layers. This approach is inspired by the power spectrum analysis of the local gravity signals, which shows that the gravity signals are superpositions of contributions from the anomaly sources at different depths, and the signals originating from different anomaly sources have heterogeneous spectral contents (Spector and Grant, 1970; Syberg, 1972; Xu et al., 2018). Since the Poisson wavelets at different depths are sensitive to signals with heterogeneous frequency characteristics, we place Poisson wavelet grids at locations where anomaly sources situate. In this manner, the contributions of the anomaly sources at various depths can be estimated.

In order to separate the contributions of different anomaly sources, the wavelet multi-scale analysis is applied to decompose the gravity data $\Delta g(\varphi, \lambda)$ into wavelet approximation $A_W(\varphi, \lambda)$ and a number of wavelet details $D_w(\varphi, \lambda)$ $(w = 1, 2, 3, \cdots, W)$ at different scales (Jiang et al., 2012; Audet, 2013; Xu et al., 2017).

$$\Delta g(\varphi, \lambda) = A_W(\varphi, \lambda) + \sum_{w=1}^{W} D_w(\varphi, \lambda) \tag{4}$$

where $(\varphi, \lambda)$ is the geodetic latitude and longitude, $W$ is the maximum order for decomposition, $A_W(\varphi, \lambda)$ is the regional anomaly caused by deep and large-scale geological bodies, and $D_w(\varphi, \lambda)$ is the local anomaly originating from shallow and small-scale heterogeneous substances. Wavelet analysis generates low-order wavelet details that are constant despite the decomposition order; only high-order wavelet details and the corresponding wavelet approximation change with decomposition order. Based on this, we can choose the proper decomposition order to obtain desirable solutions.

According to the solution to the two-dimensional Lapalace's equation, each $D_w(\varphi, \lambda)$ in eq. (4) can be expressed as (Spector and Grant, 1970; Syberg, 1972; Cianciara and Marcak, 1976)

$$D_w(\varphi, \lambda) = \sum_{\varphi} \sum_{\lambda} G_K e^{i2\pi(K_\varphi \varphi + K_\lambda \lambda)} e^{2\pi KH} \tag{5}$$

where $G_K$ denotes the amplitude, $K = \sqrt{K_\varphi^{\,2} + K_\lambda^{\,2}}$ is the wave number, and $H$ is the elevation of $D_w(\varphi, \lambda)$.

Thus, $G_K$ can be determined as:

$$G_K = \sum_\varphi \sum_\lambda D_w(\varphi, \lambda) e^{-i2\pi(K_\varphi \varphi + K_\lambda \lambda)} e^{\pm 2\pi KH} \tag{6}$$

When $H = 0$, eq. (6) can be written as

$$(G_K)_0 = \sum_\varphi \sum_\lambda D_w(\varphi, \lambda) e^{-i2\pi(K_\varphi \varphi + K_\lambda \lambda)} \tag{7}$$

Inserting eq. (7) into eq. (6), $G_K$ is rewritten as

$$G_K = (G_K)_0 e^{\pm 2\pi KH} \tag{8}$$

Hence,

$$P_K = (P_K)_0 e^{\pm 4\pi KH} \tag{9}$$

where $P_K = (G_K)^2$ is the power. Then,

$$\ln P_K = \ln(P_K)_0 \pm 4\pi KH \tag{10}$$

where $\ln P_K$ is the natural logarithm of $P_K$. Based on the linear correlation between $K$ and $\ln P_K$ in eq. (10), the corresponding average source depth $h_w$ of $D_w(\varphi, \lambda)$ can be estimated as (Spector and Grant, 1970; Xu et al., 2018)

$$h_w = \frac{1}{4\pi} \frac{\Delta \ln P_K^w}{\Delta K_w} \quad w = 1, 2, \cdots, W \tag{11}$$

where $\Delta \ln P_K^w$ and $\Delta K_w$ are the change rates of $\ln P_K^w$ and $K_w$, respectively. In this manner, the corresponding average source depths $h_w \, (w = 1, 2, \cdots, W)$ of all decomposed wavelet details $D_w(\varphi, \lambda) \, (w = 1, 2, \cdots, W)$ and wavelet approximation $A_W(\varphi, \lambda)$ can be estimated.

In this study, terrestrial and shipborne gravity anomalies are merged for modeling. Gravity anomalies, $\Delta g$, and

quasi-geoid heights, $\zeta$, are related to the disturbing potential based on the multilayer approach as follows:

$$
\begin{aligned}
\Delta g(z) &\approx -\frac{2}{|z|} T_{res}(z) - \frac{\partial T_{res}(z)}{\partial |z|} \\
&= \sum_{i=1}^{I} \sum_{n=1}^{N_i} \beta_{i,n} \left( -\frac{\partial}{\partial |z|} \Psi_i(z, y_{i,n}) - \frac{2}{|z|} \Psi_i(z, y_{i,n}) \right) \\
\zeta(z) &= \frac{T_{res}(z)}{\gamma(z)} = \sum_{i=1}^{I} \sum_{n=1}^{N_i} \beta_{i,n} \frac{\Psi_i(z, y_{i,n})}{\gamma(z)}
\end{aligned}
\tag{12}
$$

where $\gamma$ is the normal gravity value.

We assume the observational errors to be white noise with zero mean, and the gravity field model using the multilayer

approach is expressed as the standard Gauss-Markov model

$$
l_j - e_j = A_j x, \; E\{e_j\} = 0, \; D\{e_j\} = C_j = \sigma_j^2 Q_j = \sigma_j^2 P_j^{-1}, \; j = 1, 2, \cdots, J
\tag{13}
$$

where $l_j$ is the $m_j \times 1$ corresponding observation vector of group $j$, $e_j$ is the $m_j \times 1$ vector of observational errors,

$A_j$ is the $m_j \times N$ design matrix of group $j$, $x$ is the $N \times 1$ vector of unknown coefficients, including the unknown

parameters of Poisson wavelets of all the layers, i.e.

$x = \left[ \beta_{1,1}, \beta_{1,2}, \cdots, \beta_{1,N_1}, \beta_{2,1}, \beta_{2,2}, \cdots, \beta_{2,N_2}, \cdots, \beta_{I,1}, \beta_{I,2}, \cdots, \beta_{I,N_I} \right]'$, and $N = N_1 + N_2 + \cdots + N_I$; $m_j$ is the

number of observations in group $j$, and $J$ is the number of observation groups. $E\{\cdot\}$ and $D\{\cdot\}$ are the expectation

and dispersion operators, respectively. $C_j$ is the error variance-covariance matrix of group $j$, and $\sigma_j^2, Q_j$ and $P_j$ are

the variance factor, cofactor matrix, and weight matrix of group $j$, respectively.

The data in different groups are assumed to be independent, and the weight matrix $P_j$ is assumed as the scaled diagonal

matrix with white noise properties since it is usually difficult to acquire a realistic full error variance-covariance matrix in real-life measurements. Point-wise data can be directly combined for modeling through the functions described above. However, the heterogeneous characteristics of the data, in terms of spatial coverages and noise properties, may result in an ill-conditioned normal matrix (Panet et al., 2011). We apply the first-order Tikhonov regularization for

tackling the problem of the ill-conditioned matrix (Kusche and Klees, 2002; Wu et al., 2017a). For a given $\alpha$ (regularization parameter) and $\kappa$ (regularization matrix), the least-squares solution of eq. (13) is (Klees et al., 2008):

$$\hat{\boldsymbol{x}} = \left( \sum_{j=1}^{J} \left( \frac{1}{\sigma_j^2} \boldsymbol{A}_j^T \boldsymbol{P}_j \boldsymbol{A}_j \right) + \alpha \kappa \right)^{-1} \left( \sum_{j=1}^{J} \left( \frac{1}{\sigma_j^2} \boldsymbol{A}_j^T \boldsymbol{P}_j \boldsymbol{l}_j \right) \right) \tag{14}$$

Furthermore, we use Monte-Carlo variance component estimation (MCVCE) to estimate the appropriate variance factors for different observation groups and the regularization parameter (Koch and Kusche, 2002; Kusche, 2003; Wu et al., 2017c).

## 3. Numerical results and discussion

The network design of the multilayer model contains several key parameters such as the number of layers, the depth of

15 each layer, and the number of Poisson wavelets in each layer. Since the different layers are linked to the anomaly sources located at different depths, wavelet decomposition is used to separate and extract the contributions of the different anomaly sources. Moreover, the signal analysis is applied to determine the number of multiple layers based on background knowledge of local gravity field signals, while the power spectrum analysis is used to estimate the average depth of each layer. Then, we use a trial and error approach to determine the optimal number of Poisson

wavelets in each layer. A flowchart representing the design of the multilayer model is shown in Figure 1, and the details will be discussed in Sections 3.1 and 3.2.

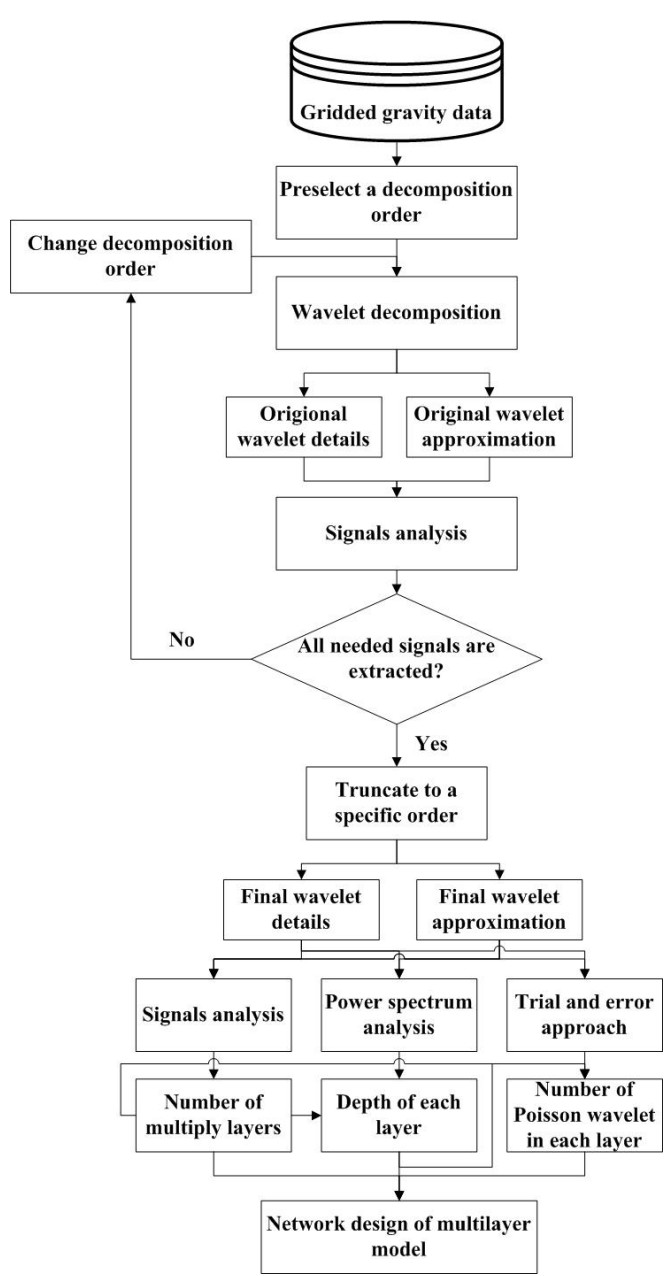

Figure 1 Flowchart showing the design of the multilayer model

## 3.1. Wavelet analysis of local gravity signals

To determine the depths of the different layers, the residual gravity data are decomposed into signals at the different

scales based on wavelet analysis. Spline interpolation is used to compute the gridded data, and Coif3 basis functions

are chosen for wavelet decomposition (Xu et al., 2017). The preliminary maximum order for wavelet decomposition is

arbitrarily chosen to some extent. However, since low-order wavelet details are constant despite change in the decomposition order, we can preliminarily choose a predefined order and implement wavelet decomposition; we then analyze the derived details in order to choose the optimal order. For signals useful for constructing the multilayer model that remain unseparated, we change the decomposition order until all the useful signals have been extracted.

Otherwise we truncate to a specific order and compute the wavelet details and approximation to conduct the multiple layers network design. By trial and error, the preliminary order for decomposition is chosen as nine. Figure 2 shows the derived wavelet details (the corresponding statistics are provided in Table 1). With the increase in the decomposition order, more long-wavelength features occur. Specifically, the low-order wavelet details indicate high-frequency signals stemming from shallow and small-scale substances, while high-order wavelet details with long-wavelength patterns

reflect anomalies caused by deep and large-scale geological bodies. The 1st- and 2nd-order details (i.e. $D_1$ and $D_2$)

seem to be dominated by high-frequency signals that correlate strongly with the local topography (the local digital terrain model (DTM) can be seen in Figure 1 in Wu et al. (2017c)). We attribute this to the uncorrected topographical signals in the RTM corrections; these remain due to the inaccuracy of the density parameters in RTM and the limitations of DTM in terms of spatial resolution and precision. As a result, high-frequency signals originating from

local topographical variations cannot be thoroughly recovered from RTM reduction, and consequently, the uncorrected signals leak into the 1st- and 2nd-order details. To avoid these high-frequency errors propagating into the final solution, we neglect these two wavelet details in designing the network of multilayer model. With nine layers, we observe $D_9$

reveal large-scale signals with wavelengths of hundreds of kilometers. Given that the mean distance between measured gravity data in this target area is approximately 6~7 km and the spatial resolution of the applied GGM (i.e. GOCO05S)

is roughly 72 km, the spectral contents of the residual signals to be recovered is roughly between several kilometers and tens of kilometers within the RCR framework, i.e. approximately between degree 250 to 3000 in terms of spherical harmonics representation. The spectral contents of the 9th-order wavelet details exceed the frequency bands of the signals to be modeled, thus, the maximum order for wavelet decomposition is truncated to eight. In this manner, the third- to eighth-order ( $D_3$ - $D_8$ ) wavelet details and the final approximation ( $A_8$ ) (see the information in Figure 3 and

Table 2) are applied for constructing the multilayer model; the model then consists of seven layers at various depths.

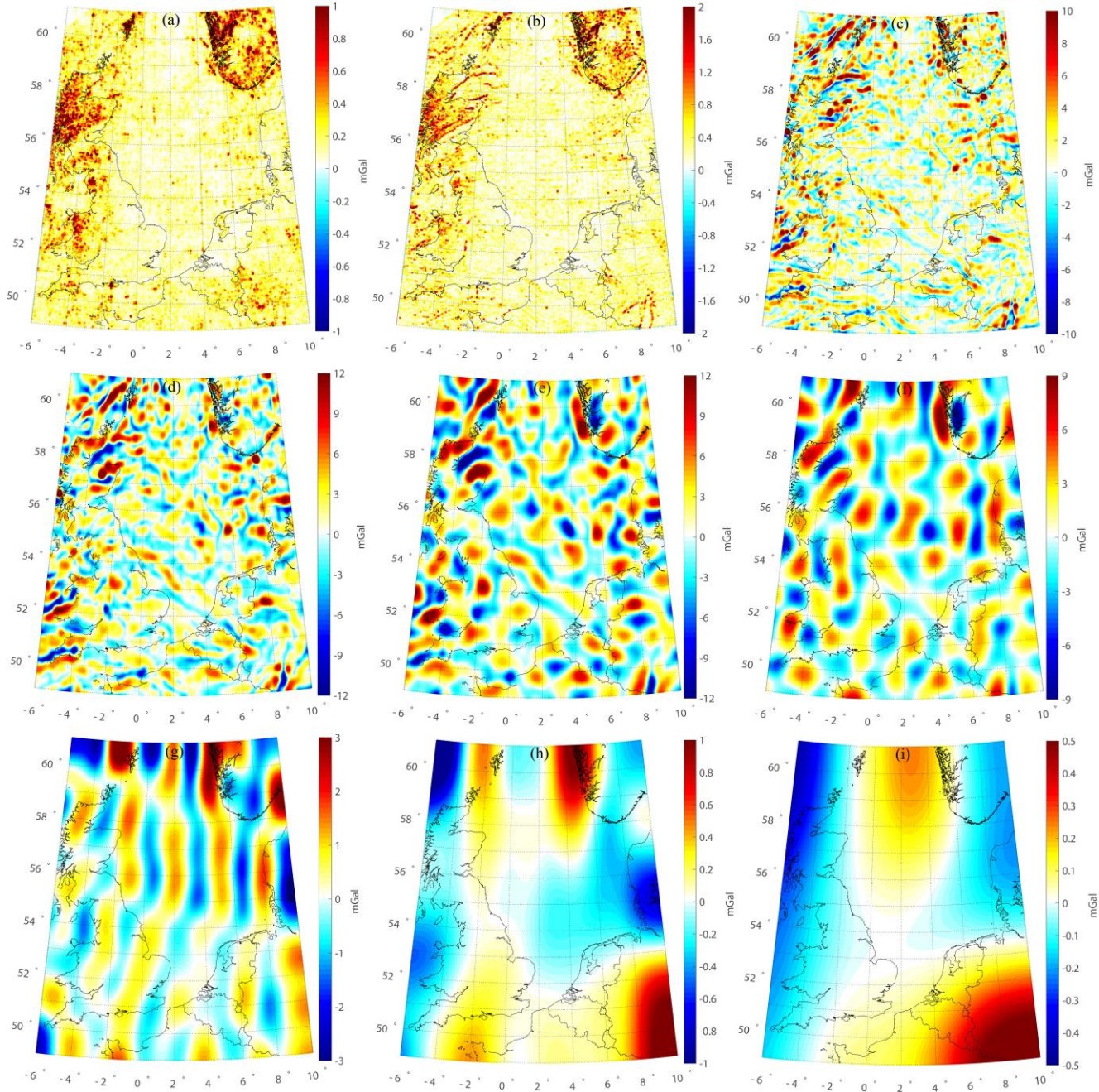

Figure 2. Wavelet details at various scales. (a) $D_1$, (b) $D_2$, (c) $D_3$, (d) $D_4$, (e) $D_5$, (f) $D_6$, (g) $D_7$, (h) $D_8$, and (i) $D_9$.

Table 1. Statistics of different wavelet details (units: mGal)

|       | Max   | Min    | Mean | Sd   |
| ----- | ----- | ------ | ---- | ---- |
| $D_1$ | 2.23  | -2.78  | 0.00 | 0.20 |
| $D_2$ | 4.52  | -5.57  | 0.00 | 0.32 |
| $D_3$ | 19.27 | -16.26 | 0.00 | 2.30 |
| $D_4$ | 21.71 | -17.46 | 0.00 | 3.18 |
| $D_5$ | 15.38 | -16.47 | 0.00 | 3.80 |
| $D_6$ | 10.60 | -9.72  | 0.00 | 2.75 |
| $D_7$ | 4.43  | -3.33  | 0.00 | 0.95 |
| $D_8$ | 1.23  | -1.52  | 0.00 | 0.34 |
| $D_9$ | 0.66  | -0.45  | 0.00 | 0.18 |

Figure 3. Wavelet approximation $A_8$.

Table 2. Statistics of wavelet approximation (units: mGal).

| Max | Min | Mean | SD |
| --- | --- | --- | --- |
| 0.83 | -1.70 | -0.41 | 0.32 |

## 3.2. Key parameters of Poisson wavelets

The order of Poisson wavelets is fixed at 3 to achieve a good compromise between localization in the space and frequency domains (Panet et al., 2011). In addition, the depth and number of Poisson wavelets are crucial factors affecting the quality of the regional solution (Klees et al., 2008). Poisson wavelets belonging to different layers are placed on the Fibonacci grids at various depths beneath the Earth's surface, and the power spectrum analysis is applied to estimate the depths. In Figure 4, the green curves show the radially averaged logarithm power spectrums of signals at different scales, and the red straight lines represent the slopes of the spectrums, indicating the depths of the corresponding layers. The red lines represent the rates of change for logarithmic power relative to the wave number, estimated by an autoregressive method. The initial and terminal points of the red lines are the inflection points of the curves, recognized according to the trend of the curves (Xu et al., 2018). Table 3 provides the estimated depths of the different layers, limited between 4 and 60 km. The shallowest layer is located 4.5 km underneath the Earth's surface, while the deepest layer is estimated to be approximately 59.2 km below the Earth's surface. The thickness of the sediments in the study area is approximately 2~4 km, and the thickness of the upper-middle crust is roughly 15~20 km (Artemieva and Thybo, 2013). Thus, the first four layers (layer 1, layer 2, layer 3, and layer 4) are located between the sediments and upper-middle crust. The corresponding wavelet details ($D_3$, $D_4$, $D_5$, and $D_6$) comprise small-scale patterns due to the highly heterogeneous structure of the crust. $D_3$ and $D_4$ correspond to the tectonic structure in the upper crust. The distributions of $D_3$ and $D_4$ (at the average depths of 4.5 km and 9.2 km, respectively) on land are more dispersed than that in the ocean, and the tectonic structure underneath the land is found to be more complex than that beneath the ocean in the upper crust. Moreover, the gravity anomalies in the northern part of North Sea are more dispersed than those in the central and southern parts of North Sea, which is consistent with the fact that the Viking Graben and two basins (i.e. Forth Approaches Basin and Norwegian-Danish Basin) are located in the northern and southern parts of North Sea, respectively (e.g. see Fichler and Hospers (1990), and Blundell et al. (1991)). The mean source depths of $D_5$ and $D_6$ are 13.7 km and 19.6 km, respectively; they correspond to the depth of the middle

crust. The gravity anomalies in these two layers present apparent positive-negative alternating patterns, which may be interpreted as the crustal shearing and extrusion (Blundell et al., 1991; Ziegler and Dèzes, 2006). The last three layers (layer 5, layer 6, and layer 7) are assumed to be located between the Moho surface and upper mantle, considering that the Moho depth in the region is approximately 25~30 km (Grad and Tiira, 2009), and the corresponding details ($D_7$, $D_8$, and $A_8$) become smoother and more long-wavelength signals occur. $D_7$, with a mean source depth of 27.0 km primarily reflects the Moho undulation. The distribution of positive-negative alternating gravity anomalies in $D_7$ is nearly south-north oriented, which is in agreement with the features of the Moho relief in the area (Fichler and Hospers, 1990; Ziegler and Dèzes, 2006). The average source depths of $D_8$ and $A_8$ are 32.3 km and 59.0 km, respectively, corresponding to the depth of the upper mantle; this indicates that the density distribution of the upper mantle is relatively smooth. Overall, these decomposed gravity anomalies can reveal the tectonic structure of the study area at different depths.

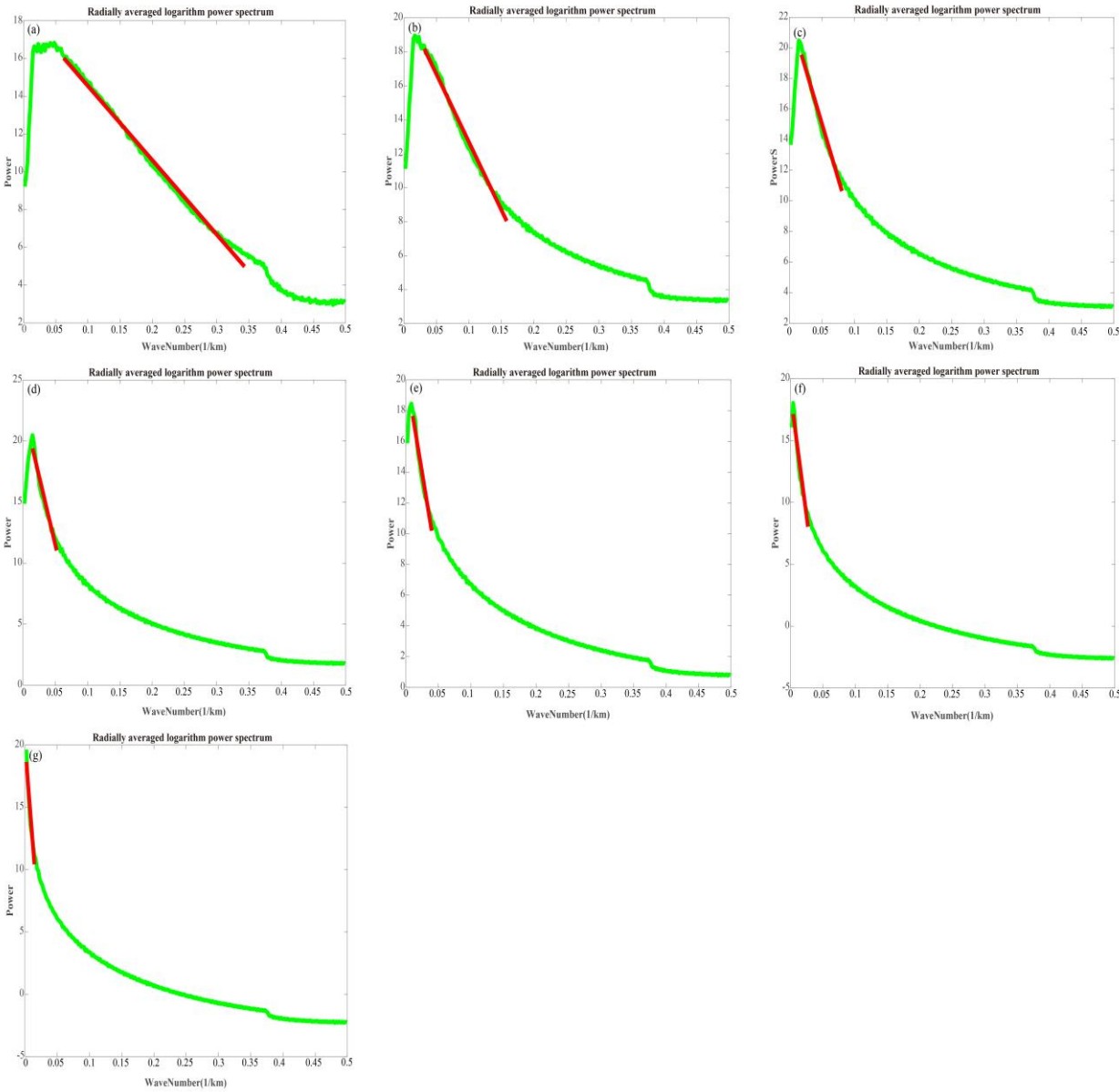

Figure 4. Power spectrum analysis of various wavelet signals. (a) $D_3$, (b) $D_4$, (c) $D_5$, (d) $D_6$, (e) $D_7$, (f) $D_8$, and (g) $A_8$.

The green curves are the radially averaged logarithm power spectrums, and the red straight lines represent the rates of change for logarithmic power relative to wave number.

Table 3 Depths of the multiple layers beneath the Earth's surface (Units: km).

| layer 1 | 4.5 |
|---|---|
| layer 2 | 9.2 |
| layer 3 | 13.7 |
| layer 4 | 19.6 |
| layer 5 | 27.0 |
| layer 6 | 32.3 |
| layer 7 | 59.2 |

A trial-and-error approach is used to estimate the number of Poisson wavelets at each layer (Wittwer, 2009). For a specific layer with a fixed depth, we predefine different numbers of Poisson wavelets to form a certain number of Fibonacci grids. Then, the signals reconstructed from these grids are compared with the true values, i.e. those derived from wavelet decomposition. The parameter that obtains the smallest difference between the modeled and true signals is considered as the optimal parameter. By trial and error, the spatial resolutions of the Fibonacci grids (mean distance between Poisson wavelets) are changed from 20 to 14 km with a step of 1 km. Table 4 shows the accuracies of the solutions derived from the different Fibonacci grids of the multiple layers, and we take the situation of the first layer for instance. With increase in the number of Poisson wavelets, the standard deviation of the differences between the reconstructed and true signals decreases gradually to 0.12 mGal when the spatial resolution of the grid increases to 16 km. Beyond this point, no significant changes occur on incorporating more Poisson wavelets. Moreover, introducing more Poisson wavelets increases the overlapping between them, which may lead to highly-conditioned normal matrices, and the associated heavy regularization may decrease the solution quality (Wu et al., 2017b). The optimal mean distance between Poisson wavelets of the first layer is estimated as 16 km. Similarly, the spatial resolutions for the remaining layers can be determined in this way (see Table 4).

Table 4 Accuracies of solutions derived from the various Fibonacci grids for different layers (Units: mGal).

| | 20 km | 19 km | 18 km | 17 km | 16 km | 15 km | 14 km |
|---|---|---|---|---|---|---|---|

| | | | | | | | |
|---|---|---|---|---|---|---|---|
| layer 1 | 0.43 | 0.34 | 0.21 | 0.16 | 0.12 | 0.12 | 0.12 |
| layer 2 | 0.52 | 0.43 | 0.33 | 0.25 | 0.19 | 0.16 | 0.16 |
| layer 3 | 0.58 | 0.40 | 0.28 | 0.19 | 0.16 | 0.14 | 0.14 |
| layer 4 | 0.55 | 0.39 | 0.29 | 0.26 | 0.15 | 0.13 | 0.13 |
| layer 5 | 0.38 | 0.26 | 0.17 | 0.14 | 0.10 | 0.10 | 0.10 |
| layer 6 | 0.22 | 0.16 | 0.12 | 0.10 | 0.08 | 0.08 | 0.08 |
| layer 7 | 0.11 | 0.09 | 0.08 | 0.06 | 0.06 | 0.06 | 0.06 |

## 3.3. Regional solution and its validation

For regional gravity field recovery, point-wise terrestrial and shipborne gravity anomalies are combined. Since there is no accurate information on the accuracies of terrestrial and shipborne data, we assume an accuracy of 2 mGal for both of these types of data, and the posterior variance factors of different data are estimated from MCVCE. The weights of different data indicate their relative contributions, and play a key role in data combination. The estimated variance factors for terrestrial and shipborne gravity data are approximately 1.45 mGal and 1.30 mGal, respectively, when we model the local gravity field based on the multilayer approach. For terrestrial data, the estimated accuracy agrees with that derived by Klees et al. (2008), i.e. 1.48 mGal for parts of the Netherlands. However, it is difficult to judge whether this estimate is realistic in other regions, because of a lack of accuracy information. For shipborne data, the computed value of 1.30 mGal is smaller than the results of crossover adjustments, where the standard deviation for the residuals at the crossovers was estimated to be approximately 2.0 mGal (Slobbe, 2013). However, this value may be too optimistic, considering that much of the shipborne data were collected decades ago without GPS navigation. The first-order Tikhonov regularization is used to tackle the ill-conditioned problem (Kusche and Klees, 2002; Wu et al., 2017b), and the convergent regularization parameter is estimated to be approximately $0.5 \times 10^{-5}$ using the MCVCE method. Details on regularization parameter estimation and comparisons with different methods can be referred to Wu et al. (2017b).

The performance of the single-layer approach is also investigated for comparison. The parameterization of the local gravity field based on the single-layer approach has been described in detail by e.g. Klees et al. (2008) and Slobbe (2013). By trial and error, the single layer of the Poisson wavelets' grid is found to be located 40 km beneath the

Earth's surface, and the mean distance between the Poisson wavelets is defined as 8.7 km (Wu et al., 2016b). Figure 5 shows the normalized spectrums for different approaches. Considering the frequency range of the signals to be recovered in the target area is approximately between degree 250 to 3000 in spherical harmonics representation, we note the single-layer approach is only sensitive to a part of the signal spectrum. It is sensitive approximately between degree 300 to 1200 if we assume the criterion for determining whether it is sensitive or not within a specific frequency band to be half of the maximum value of the normalized spectrum. However, for the high-frequency band between degree 1200 to 3000, this approach is less sensitive. On the contrary, the multilayer approach effectively covers the spectrum of the local gravity signals, and is sensitive to both the low- and high-frequency bands. The residuals of data after least squares adjustment using different methods are displayed in Figure 6 (the boundary limits for this area are contracted by 0.5 ° in all the directions to reduce edge effects).The residuals derived from the multilayer approach are significantly lower throughout the region compared with those obtained from the single-layer approach, especially in the western parts of UK, south of Norway, and southwest parts of Germany. In these regions high-frequency signals that are correlated with local topography dominate the regional gravity field. We also note improvements in the oceanic parts, especially in the waters around the English Channel, Irish Sea, northwest of the North Sea, and the Atlantic Ocean close to northwest UK. Table 5 displays the standard deviation (SD) value for the residuals of terrestrial (shipborne) gravity anomalies, which decreases by 0.37 mGal (0.34 mGal) when the multilayer approach is used. These results are reasonable, since the multilayer model contains several layers shallower than 40 km, and the spectrums of these layers shift to the high-frequency bands. Thus, the spectrum of the multilayer approach is more sensitive to high-frequency signals, and consequently, the local high-frequency signals can be better fitted by the multilayer approach. It is also worth mentioning that the analysis of data residuals cannot be treated as the only criterion to justify the performances of different approaches. There are two major reasons for this. First, these gravity data have been used for modeling purposes, and the SD values of data residuals should be regarded as the internal agreement. Additionally, due to the limitation of the accuracies of gravity data, we cannot arrive at firm conclusions based only on the analysis of data residuals. It is also possible that lower data residuals can be derived if we place the Poisson wavelets' grid at a shallower depth when the single-layer approach is used. However, we think a shallower single grid may reduce the data residuals, but may not derive a better solution when validated against the independent control data, as described in detail by Wu et al. (2016b). In the following part, we introduce another high-quality independent data set for external validation, i.e. GPS/leveling data, which gives us more confidences with respect to the performances of different methods.

It is also of interest to implement an Akaike information criterion (AIC) test for different models. Although, the multilayer model fits the gravity observations better, it also increases the level of estimated parameters. AIC rewards the goodness of fit of data, but also includes a penalty as the number of estimated parameters increases. In other words,

it deals with the trade-off between the goodness of fit of the data and the simplicity of the model. The AIC value is an estimator of the relative quality of statistical models for a given set of data, providing a means for model selection. The model that yields the minimum AIC value may be more preferable (Akaike, 1974; Burnham and Anderson, 2002). The definition for the AIC value can be seen in eq. ( 1 ) in the Appendix. Since we model the gravity field in the framework of least squares system, we can simply take $AIC = 2k + m \ln(RSS / m)$ for model comparison, where $k$ is the number of

estimated parameters in the model, $m$ is the number of observations, and $RSS$ is the residual sum of squares (RSS). For details, see the Appendix. In this study, 894649 point-wise gravity observations are used for modeling, and 47504 and 19477 parameters are estimated in the multilayer and single-layer models, respectively. The RSS values for the multilayer and single-layer model are computed as $8.8527 \times 10^5 \ mGal^2$ and $1.3296 \times 10^6 \ mGal^2$, respectively, based on the data residuals after the least squares adjustment. Then, the AIC values for the multilayer and single-layer model

are estimated as 85581 and 393400, respectively. Based on these statistics, we note that the multilayer model yields a smaller AIC value, which may be more preferable because it achieves a better balance between the goodness of fit of the data and the simplicity of the model.

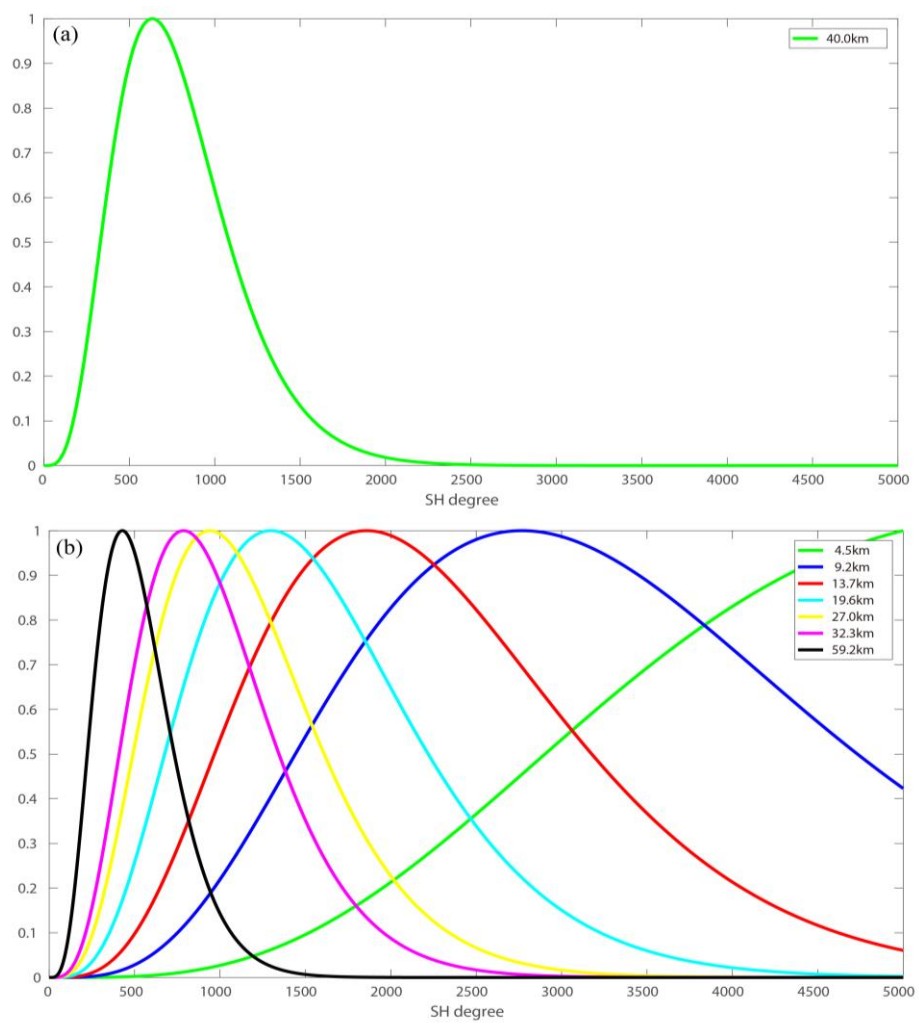

Figure 5. Normalized spectrum for (a) single-layer and (b) multilayer approach.

To test the ability of realistic extrapolation of different regional models recovered from various methods, we introduce
GPS/leveling data in the Netherlands (534 points), Belgium (2707 points), and parts of Germany (213 points) as the
independent validation data. This is a comparison of the predicted values derived from the regional model (e.g. model
computed from the multilayer or single-layer approach) and those derived from independent survey/measurements.
These data were provided in terms of geometric quasi-geoid heights derived from the high-quality GPS measurements
and leveling surveys. The overall estimated accuracy of these observed quasi-geoid heights was approximately at the 1
cm level. It is worth mentioning that these GPS/leveling data have not been combined for modeling, and their

three-dimensional coordinates do not coincide with the positions of gravity data. For validating purposes, it is necessary to reconstruct the regional model based on the estimated Poisson wavelets' coefficients and coordinates of GPS/leveling points (see Eq.(12)), and compute the gravimetric quasi-geoid heights at these predicted points. We compute the standard deviation of the point-wise difference between GPS/leveling data and the gravimetric quasi-geoid height derived from the regional approach. This serves as an external validation.

The validation results demonstrate that the discrepancies between the GPS/leveling points and quasi-geoid heights derived from the multilayer approach decrease substantially compared with those computed from the single-layer approach (Figure 7). The most prominent improvements occur in the northwest of Belgium, west of Germany, and eastern parts of Netherlands, which are in good agreement with the results for data residuals analysis demonstrated in Figure 6. As shown in Table 6, the accuracies of gravimetric quasi-geoid derived from the multilayer approach improve by 0.4 cm, 0.9 cm, and 1.1 cm in the Netherlands, Belgium, and parts of Germany, respectively. Moreover, the mean values indicate that the solution computed from the multilayer approach further reduces the biases between the gravimetric solution and local GPS/leveling data, with magnitudes of 0.8 cm, 0.7 cm, and 1.1 cm in these three regions, respectively, compared to the those modeled from the single-layer approach. From these results, we can see that the multilayer approach not only leads to a reduction for the data residuals, but also generates a better solution assessed by the independent control data. To construct the multilayer model, we consider that the gravity signals are the sum of the contributions generated from the anomaly sources, and different layers are designed for recovering these contributions with heterogeneous spectral contents. As a result, the spectrum of the multilayer approach is sensitive to the frequency bands of local gravity signals, both in the low- and high-frequency bands, and the local signals may be better recovered. We also notice that there are still biases between the regional gravimetric solutions and local GPS/leveling data (see the mean values in Table 6), which are mainly due to the commission errors in the GGM and uncorrected systematic errors in the local gravity data and leveling systems (Fotopoulos, 2005). Generally, corrector-surface (Fotopoulos, 2005; Nahavandchi and Soltanpour, 2006) or more complicated algorithms, like least squares collocation (Tscherning, 1978), boundary-value methodology (Klees and Prutkin, 2008; Prutkin and Klees, 2008), and a direct approach (Wu et al., 2017a), can be applied to reduce the systematic errors and properly combine GPS/leveling data and gravimetric solutions. However, since the objective of this study is to develop a multilayer approach for gravimetric quasi-geoid modeling that may be served as a basis for further geophysical applications, the derived quasi-geoid is not purely gravimetric with implementing the data merging approach. Furthermore, we only have well distributed GPS/leveling

data in a limited region, i.e. in the Netherlands, Belgium, and parts of Germany, while in other regions, no high-quality control data are available. If we use the locally distributed GPS/leveling data to remove these systematic errors and computing the combined quasi-geoid, the final solution may be distorted in other regions, especially around the ocean, because no control data exist in these regions. Thus, we do not implement the methods mentioned above for computing the combined quasi-geoid. We use the gravimetric model derived from the multilayer approach for the following study, which is hereafter denoted as QGNSea V1.0 (quasi-geoid over the North Sea version 1.0).

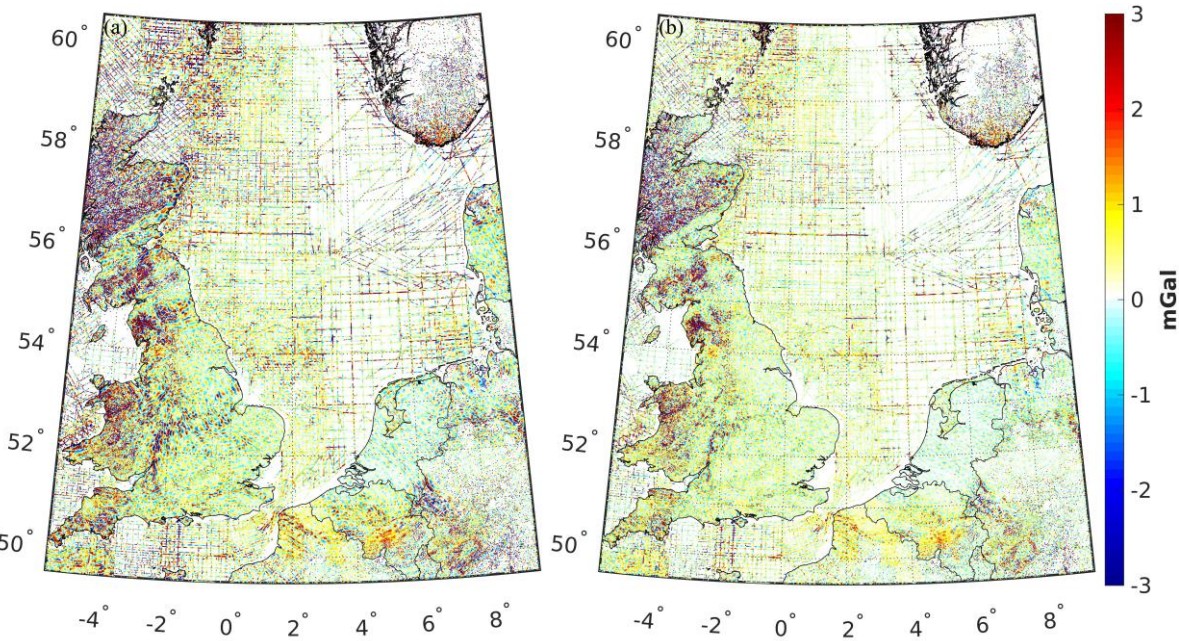

Figure 6. Residuals of gravity data derived using the (a) single-layer and (b) multilayer approach.

Table 5 Statistics of residuals of gravity data computed using different approaches (units: mGal).

|  |  | Max | Min | Mean | SD |
|---|---|---|---|---|---|
| Single-layer approach | Terrestrial | 19.58 | -16.91 | 0.00 | 1.37 |
|  | Shipborne | 11.91 | -17.38 | 0.00 | 1.02 |
| Multilayer approach | Terrestrial | 16.96 | -14.90 | 0.00 | 1.00 |
|  | Shipborne | 9.25 | -15.96 | 0.00 | 0.68 |

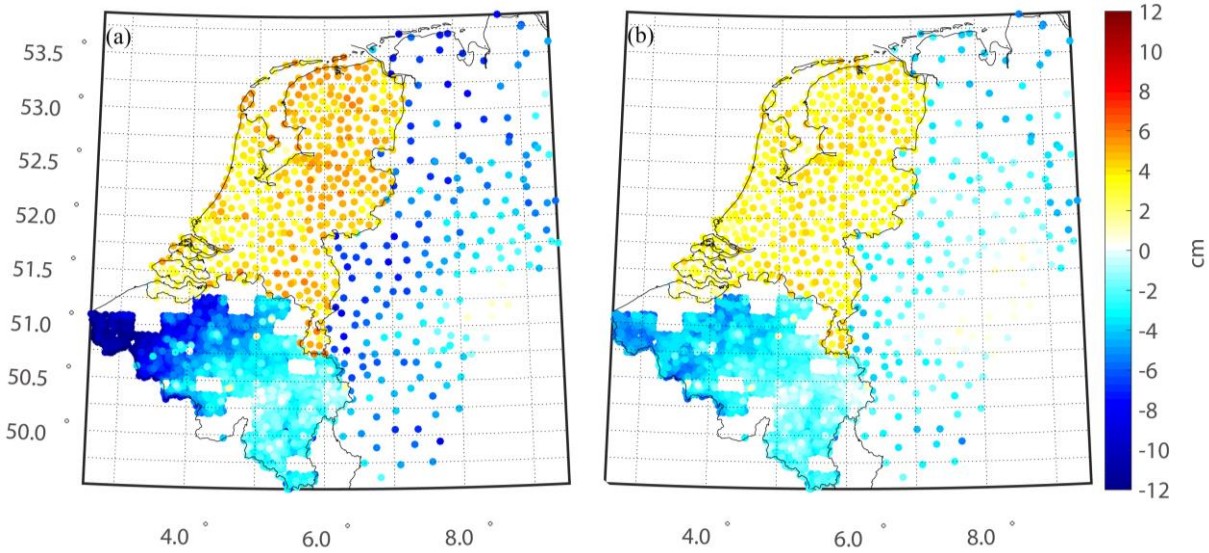

Figure 7. Differences between GPS/leveling data and gravimetric quasi-geoids computed using the (a) single-layer and (b) multilayer approach.

Table 6 Evaluation of gravimetric quasi-geoids modeled using different approaches (Units: cm).

|  |  | Max | Min | Mean | SD |
|---|---|---|---|---|---|
| Single-layer approach | Netherlands | 5.9 | 0.1 | 3.8 | 1.2 |
|  | Belgium | 1.2 | -13.1 | -3.5 | 2.8 |
|  | Germany | 1.2 | -11.2 | -3.6 | 2.9 |
| Multilayer approach | Netherlands | 4.8 | 0.0 | 3.0 | 0.8 |
|  | Belgium | 1.2 | -6.8 | -2.8 | 1.9 |
|  | Germany | 1.0 | -6.7 | -2.5 | 1.8 |

QGNSea V1.0 is compared with a regional model called EGG08 (Denker, 2013), and four other recently published high-order GGMs, i.e. EGM2008 with a full degree and order (d/o) of 2190 and 2159 (Pavlis et al., 2012), EIGEN-6C4 (d/o 2190) (Förste et al., 2014), GECO (d/o 2190) (Gilardoni et al., 2015), and SGG-UGM-1 (d/o 2159) (Liang et al., 2018). The reason for choosing these four GGMs for comparisons is that these models have relatively higher spatial resolutions and better accuracies compared to most other available GGMs (see the information in http://icgem.gfz-potsdam.de/home). EGG08 is a regional gravimetric quasi-geoid model in Europe, which was

recovered by Stokes' integral based on locally distributed gravity data. This model is provided in terms of gridded data instead of spherical harmonics, and its spatial resolution is 1′ in latitude and 1.5′ in longitude, respectively (Denker, 2013). The other four models are global geopotential models provided in terms of spherical harmonics, and EGM2008 was computed by merging GRACE measurements, terrestrial, altimetry-derived, and airborne gravity data. Since no GOCE data have been incorporated for developing EGM2008, and the recently published GGMs have been developed by combining GOCE data, which is supposed to improve the gravity field in the frequency bands approximately from degree 30 to 220 in spherical harmonics representation (Gruber et al., 2010). EIGEN-6C4 was computed by combining GRACE, GOCE, and terrestrial gravity data and other data sets; GECO was computed by incorporating the GOCE-only TIM R5 (d/o 250) solution into EGM2008, and SGG-UGM-1 was computed by the combination of EGM2008 gravity anomalies and GOCE gravity gradients and satellite-to-satellite tracking data. The differences between QGNSea V1.0 and other models are shown in Figure 8 (the boundary limits for the area are reduced by 0.5 °in all directions to reduce edge effects), the magnitude of which reaches the decimeter level. For EGG08, we note the most prominent differences appear in the eastern parts of the Irish Sea and center of Germany. Different data pre-processing procedures and methods for parameterization partly account for these differences. For example, QGNSea V1.0 is recovered from the multilayer approach using Poisson wavelets, and proper weights for different observation groups are estimated through MCVCE, while the spectral combination technique and spectral weights were implemented in EGG08 for merging heterogeneous data (Denker, 2013). Larger differences are observed between QGNSea V1.0 and these four GGMs, and remarkable differences are seen in southern Norway, northern parts of the North Sea, eastern parts of the Irish Sea, and northwest parts of Germany. These differences are interpreted as resulting from the different modeling techniques, and the additional signals introduced by QGNSea V1.0, stemming from the incorporation of more high-quality gravity data. The evaluation results with GPS/leveling data displayed in Figure 9 and Table 7 show that the gravimetric quasi-geoid inversed from the multilayer approach has the best quality, especially in the north of the Netherlands and western and eastern parts of Belgium. Note that we removed the mean values between the gravimetric model (both for the regional models and GGMs) and local GPS/leveling data, since these GGMs deviate from the local GPS/leveling data by tens of centimeters or even more in this area, due to the commission errors and uncorrected systematic errors in gravity data and inconsistencies among different height datums. Thus, if the mean biases are not removed, these differences can become dominated by the systematic errors, which is undesirable for model comparison. The SD value of the misfit between the GPS/leveling data and QGNSea V1.0 is 1.5 cm, while this value increases to 2.2 cm for EGG08. In contrast, the accuracies of the four GGMs, approximately at 2.6

cm levels, are slightly worse than that of EGG08. Compared to the GGMs, the added values introduced by local high-quality data lead to the primary improvements in QGNSea V1.0. We find that the four GGMs have comparable accuracies. However, those developed by combining GOCE data and EGM2008 (i.e. GECO and SGG-UGM-1) do not demonstrate better performances than EGM2008 alone, with SGG-UGM-1 even showing a slightly worse performance than EGM2008. This is particularly prominent in the eastern parts of Belgium. However, the possible reasons require further investigation. A new Europe gravimetric quasi-geoid model, EGG2015, is also observed to have been computed, where the GOCE-derived GGMs were used as reference models (Denker, 2015). However, this model is not publicly available, and its performance cannot be assessed in this local region. Systematic errors can be seen in the results presented in Figure 9. These errors remain because they cannot be thoroughly removed by simply removing the mean differences. However, as mentioned previously, the target of this study is to develop a multilayer approach for gravimetric quasi-geoid modeling. Implementing the data merging approach for combining local GPS/leveling and gravimetric model may lead to a distorted solution. Thus, a detailed discussion regarding the removal of these systematic errors is out of the scope of this study.

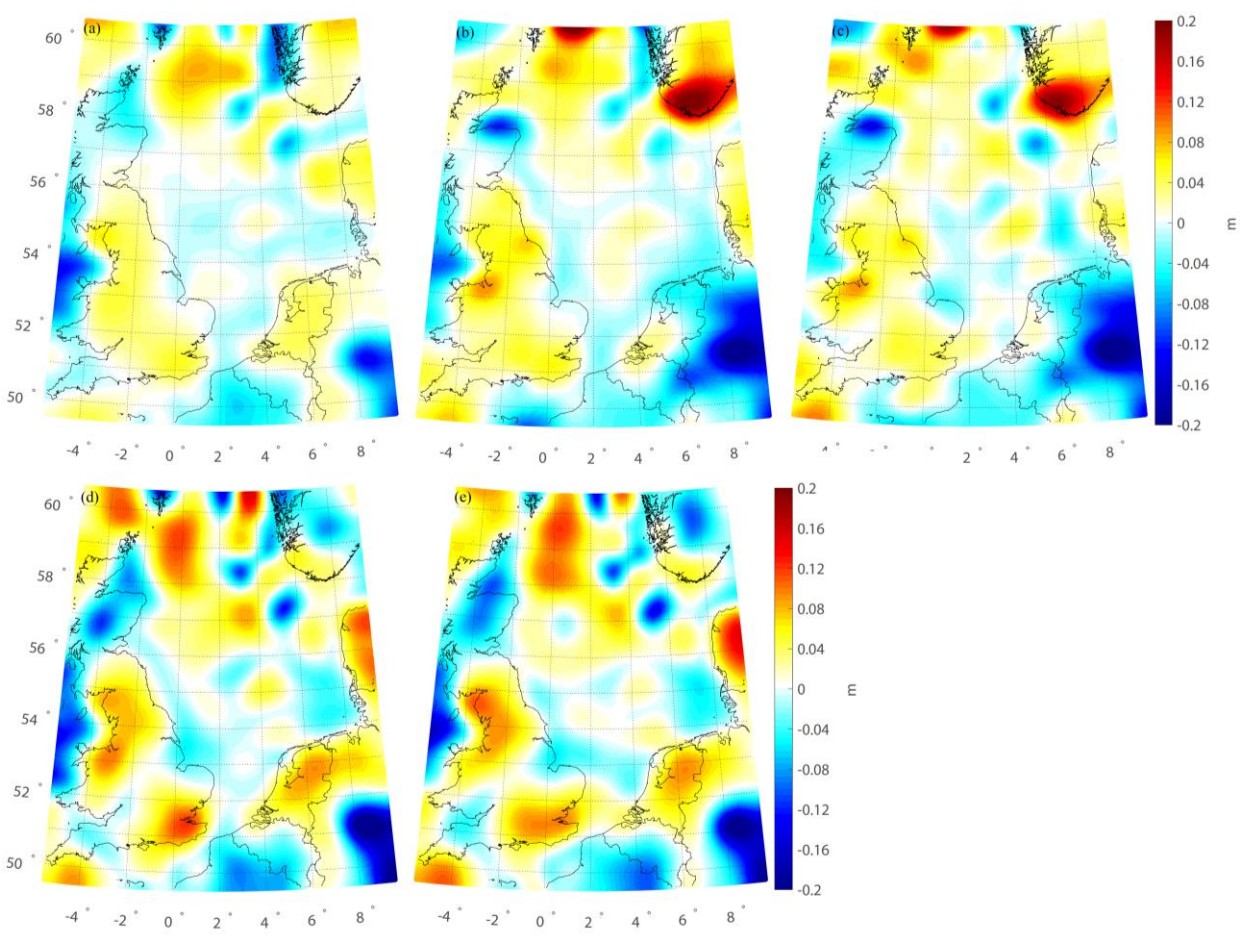

Figure 8. Difference between QGNSea V1.0 and (a) EGG08, (b) EGM2008, (c) EIGEN-6C4, (d) GECO, (e) SGG-UGM-1. Note that the mean differences are removed.

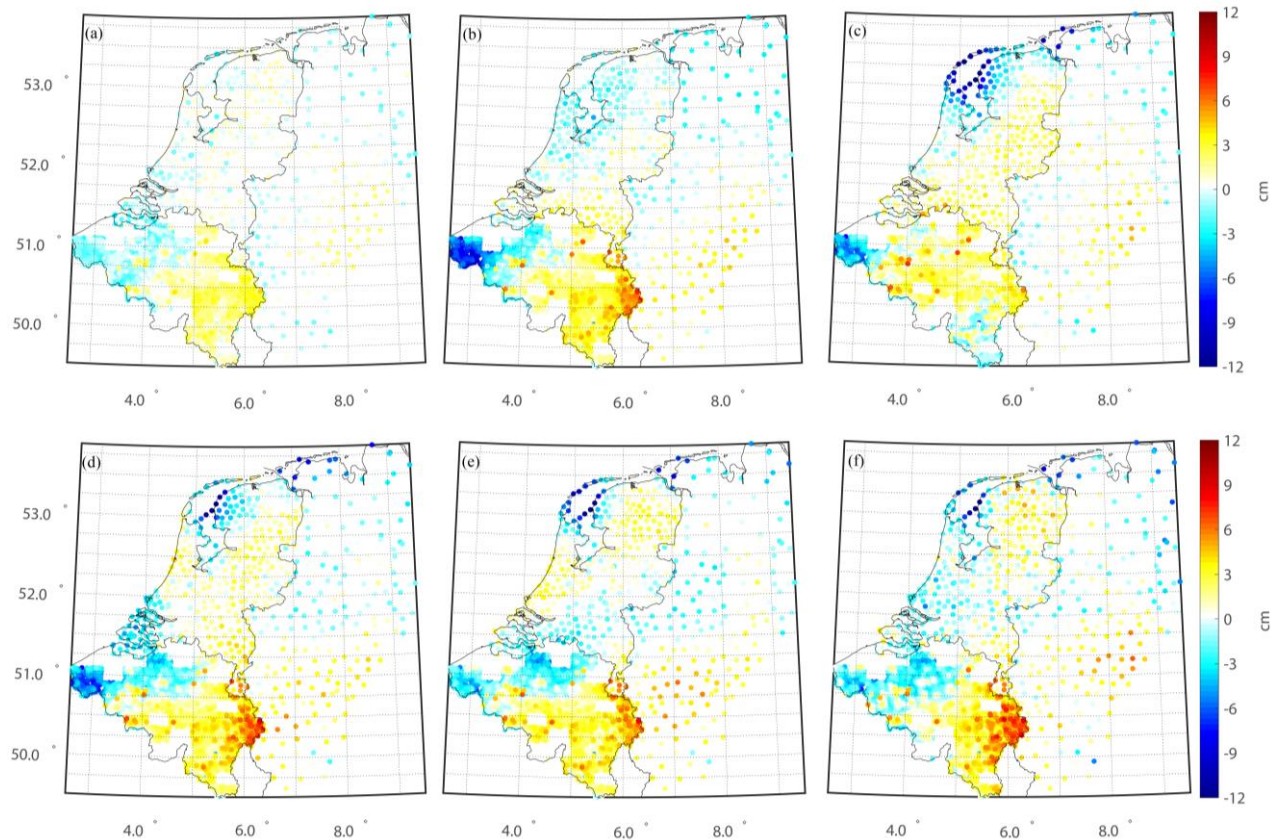

Figure 9. Evaluation of the various gravimetric quasi-geoids. (a) QGNSea V1.0, (b) EGG08, (c) EGM2008, (d) EIGEN-6C4, (e) GECO, and (f) SGG-UGM-1. Note that the mean differences are removed.

Table 7. Statistics of accuracy for various gravimetric quasi-geoids. (units: cm). Note that the mean differences are removed.

|  | Max | Min | SD |
| --- | --- | --- | --- |
| QGNSea V1.0 | 5.2 | -3.9 | 1.5 |
| EGG08 | 7.8 | -9.4 | 2.2 |
| EGM2008 | 8.4 | -10.0 | 2.6 |
| EIGEN-6C4 | 9.0 | -11.9 | 2.7 |
| GECO | 8.3 | -12.8 | 2.6 |
| SGG-UGM-1 | 8.8 | -12.7 | 2.7 |

For further comparison, we compute the local mean dynamic topography (MDT), which illustrates the departure of the mean sea surface (MSS) from the quasi-geoid/geoid (Becker et al., 2014; Bingham et al., 2014). We compute the MDTs in a geodetic manner, with raw MDTs computed as the differences between MSS and local geoid/quasi-geoid models. The derived MDTs are further smoothed with a Guassian filter to suppress the small-scale signals from the MSS or local geoid/quasi-geoid that cannot be resolved (Andersen et al., 2013). The DTU13MSS from 1993-2012 is chosen as the MSS, and this model is provided as the gridded data with a spatial resolution of $1' \times 1'$ (Andersen et al., 2013). Considering that QGNSea V1.0 and EGG08 have better performances than other models when validated against local GPS/leveling data, we only compute local MDTs based on these two gravimetric quasi-geoid models. DTU13MSS and QGNSea V1.0/EGG08 are directly combined to obtain the raw MDT. Then, a Gaussian filter with a correlation length of 6 km is further applied to smooth the derived MDT, considering the signals at very short scales can not be recovered from the local gravity data, due to the limited spatial resolution of the gravimetric measurements.

The MDTs modeled based on QGNSea V1.0 and EGG08 are denoted as MDTNS_QGNSea and MDTNS_EGG08, respectively (Figure 10). The results of these models agree with each other in most regions over the North Sea. Prominent signals such as the Norwegian coastal currents can be seen in the MDTs, e.g. see Idžanović et al. (2017). The signals observed in MDTNS_QGNSea do not provide a full picture of Norwegian coastal currents due to the limited data coverage in Norway and its neighboring ocean areas. In most areas of the North Sea, the MDTs show considerably smooth patterns, indicating a small change in the sea surface topography; this result is consistent with Hipkin et al. (2004). However, extreme values are observed surrounding most offshore areas, e.g. see the features over the offshore regions closed to The Wash (around 0.5 °W and 53 °N) and Thames estuary (around 1 °W and 51.5 °N) in England, and along the coastal areas of France, Netherlands, and Germany. MDT signals in these areas are traditionally difficult to model and are frequently identified as errors (Hipkin et al., 2004). The problems for computing geodetic MDTs in offshore regions are twofold. First, the quasi-geoid/geoid is poorly modeled in coastal areas due to unfavorable data coverage, and data inconsistencies are usually observed when combining land and marine gravity surveys. Further, the quality of altimetry data is dramatically reduced near offshore areas, and associated errors in the derived MSS propagate into the final MDT (Andersen et al., 2013). However, airborne gravimetric survey provides a seamless method of gravity measurements over land and oceans, which may improve this situation (Andersen and Knudsen, 2000).

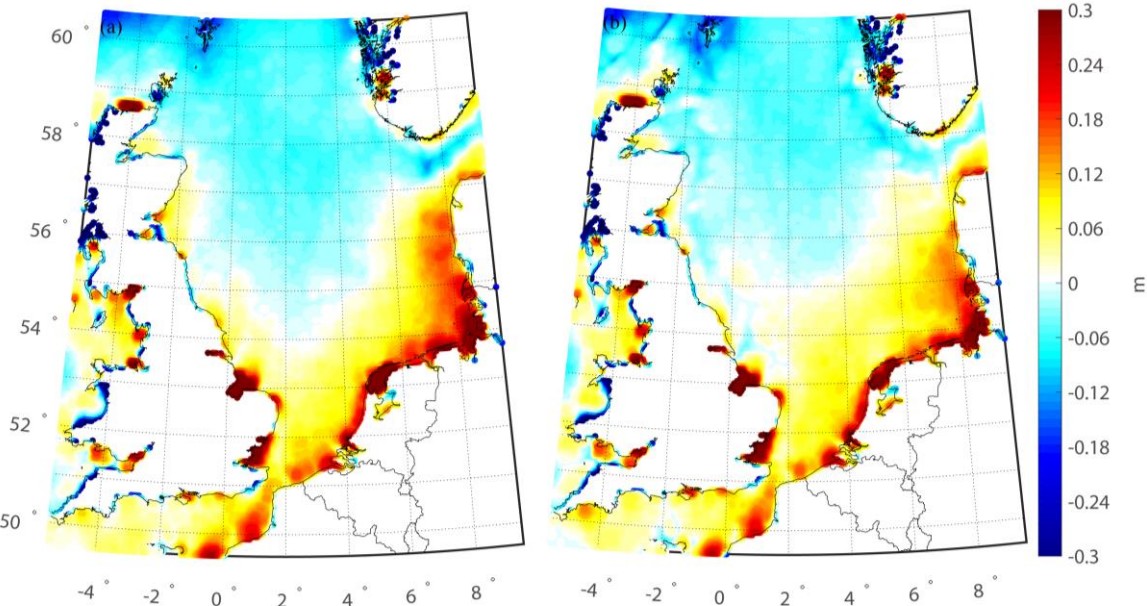

Figure 10. Different geodetic MDTs over the North Sea. (a) MDTNS_QGNSea; (b) MDTNS_EGG08. For all profiles, the mean values have been removed.

## 4.  Conclusions

A multilayer approach for gravity field recovery at the regional scale, within the framework of multi-resolution representation, is developed, where the residual gravity field is parameterized as the superposition of the multiple layers of Poisson wavelets located at different depths beneath the Earth's surface. Since the gravity signals are the sum of the contributions of the anomaly sources at different depths, we place the multiple layers of the model at the locations of the different anomaly sources. Further, wavelet decomposition and power spectrum analysis are applied to estimate the depths of the different layers.

To test the performance of this multilayer approach, we model a local gravimetric quasi-geoid model, QGNSea V1.0, over the North Sea and validate this model against independent control data. Based on wavelet decomposition and power spectrum analysis, multiple layers located between 4.5 km and 59.2 km underneath the Earth's surface are constructed to capture signals at different scales. The numerical results show that the multilayer approach is sensitive to the spectrum of signals, both in the low- and high-frequency bands, while the traditional single-layer approach is only sensitive to parts of the signals' spectrum. Comparisons with the single-layer approach show that the multilayer

approach fits the gravity observations better, especially in the regions where the gravity signals show strong correlations with the variation of local topography. Moreover, an Akaike information criterion (AIC) test, which estimates the relative quality of the statistical models for a given set of data, is introduced for model selection in view of statistical test. The associated results demonstrate that the multilayer model obtains a smaller AIC value and achieves a better balance between the goodness of fit of data and the simplicity of the model. Evaluation using independent GPS/leveling data tests the ability of regional models recovered from different methods towards realistic extrapolation, and shows that QGNSea V1.0 using the multilayer approach fits the local GPS/leveling data better than that using the single-layer approach, by the magnitudes of 0.4 cm, 0.9 cm, and 1.1 cm in the Netherlands, Belgium, and parts of Germany, respectively,. Further comparisons with the existing models show that QGNSea V1.0 is superior in terms of performance and may be beneficial for investigating ocean circulation in the North Sea and surrounding oceanic areas.

Future work should focus on further improving the QGNSea V1.0. First, a data-adaptive algorithm may be developed for designing the optimal network in the multilayer approach, such as an algorithm for choosing the order for wavelet decomposition and determining the number of multiple layers, since human interventions are currently needed for estimating these key parameters. Moreover, satellite data (e.g. K-band Range Rate data and gravity gradients) from GRACE and GOCE missions can be combined with ground-based gravimetry and altimetry data through the multilayer approach. Doing so can further improve the quality of local gravity field recovery, especially in the long-wavelength bands. However, deeper layers than those used in this study to combine surface data may be implemented to incorporate satellite observations, since these data mainly contribute to low-frequency bands of the gravity field. In addition, the stochastic model may need to be refined. For instance, the effects of the GGM errors on the solutions can be quantified if the full error variance-covariance matrix of the spherical coefficients is incorporated into the stochastic model. Thus, the different data may be more properly weighted and the solutions may be further improved.

*Author contributions.* All authors have contributed to designing the approach and writing the manuscript.

*Code and data availability.* The source code is included as the Supplement. Gravity data were provided by the British Geological Service; the Geological Survey of Northern Ireland; the Nordic Geodetic Commission; Bundesamt für Kartographie und Geodäsie (Germany); Institut für Erdmessung (Germany); the Bureau Gravimétrique International

IAG service (France); the Banque de données Gravimétriques de la France; and the Bureau de Recherches Géologiques et Minières (France). GPS/leveling data were provided by the Geo-information and ICT of Rijkswaterstaat (RWS-AGI) and the GPS Kernnet of the Kadaster, National Geographic Institute (NGI) and the Royal Observatory (ROB), and Bundesamt für Kartographie und Geodäsie.

*Competing interests.* The authors declare that they have no conflict of interest.

*Acknowledgments*. The authors would like to give our sincerest thanks to two anonymous reviewers and Dr. Cornelis Slobbe for their beneficial suggestions and comments, which are of great value for improving and correcting the manuscript. We also thank the Executive Editor Lutz Gross for the kind assistances and constructive comments. We thank the kind supports from the editorial office. We acknowledge funding from the Netherlands Vertical Reference
Frame project. Thanks Prof. Roland Klees and Dr. Cornelis Slobbe from Delft University of Technology for kindly providing the original software. This study was supported by the Fundamental Research Funds for the Central Universities (No.2018B07314), the National Natural Science Foundation of China (No.41830110, 41474061, 41504015, 41474109), the State Scholarship Fund from Chinese Scholarship Council (No.201306270014), the Open Research Fund Program of the State Key Laboratory of Geodesy and Earth's Dynamics (No.SKLGED2018-1-2-E and
SKLGED2018-1-3-E), and Key Laboratory of Geospace Environment and Geodesy, Ministry of Education, Wuhan University (No.17-01-09).

# Appendix A: Akaike information criterion

Suppose that we have a statistical model of some data, and the Akaike information criterion (AIC) value of the model is (Burnham and Anderson, 2002)

$$AIC = 2k - 2\ln(\hat{L}) \qquad (1)$$

where $k$ is the number of estimated parameters in the model, and $\hat{L}$ is the maximum value of the likelihood function for the model (Akaike, 1974; Burnham and Anderson, 2002).

For least squares fitting, the maximum likelihood estimate for the variance of a model's residuals distributions is

$$\hat{\theta}^2 = RSS / m \qquad (2)$$

where $RSS$ is the residual sum of squares (RSS), and $m$ is the number of observations.

Then, the maximum value of a log-likelihood function of least square model is (Burnham and Anderson, 2002)

$$-\frac{m}{2}\ln(2\pi) - \frac{m}{2}\ln(\hat{\theta}^2) - \frac{1}{2\hat{\theta}^2}RSS = -\frac{m}{2}\ln(RSS/m) + C \qquad (3)$$

where $C$ is a constant independent of the model.

Combining eq. (1) and eq. (3), for least square model, the AIC value is expressed as

$$AIC = 2k + m\ln(RSS/m) + C \qquad (4)$$

Since only differences in AIC are meaningful, the constant $C$ can be ignored, and we can conveniently take $AIC = 2k + m\ln(RSS/m)$ for model comparisons.

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
