# Peer review of "A multilayer approach and its application in modeling QGNSea V1.0: a local gravmetric quasi-geoid model over the North Sea"

_Geoscientific Model Development, 2017_

## Referee Comment (RC1) · Anonymous Referee #1 · 24 Jan 2018

Authors present elegant and well-written numerical study for the SRBF gravimetric quasigeoid modelling using the multi-layer approach and compared results with a single-layer approach. This case study is very suitable for geodetic proceedings, but the modelling of quasigeoid surface is out of geophysical interest. This is main reason I recommend rejection of this article. Authors attempt to add some geophysical content (page 12/ line 14 to page 13/ line 5) is irrelevant. This is also evident from gravity signal decomposition in Fig. 2 that does not reflect any real geological features, rather than reflects the properties of kernel for different depths. There are additional major issues to be addressed by authors before considering further publication.

[Figure]

1/ The values of variance factors for different types of observations are not given, so final accuracy and -most importantly - the claim that multi-layer approach provides better accuracy is not justified. This is especially evident from Table 5, where achieved accuracy in terms of gravity residuals is much too optimistic, because errors of gravity observations (especially for ship-borne data) are larger.

2/ Another aspect related to validation of results is the ability of realistically extrapolating the gravity field. For this purpose sets of control point is chosen with given values that are not incorporated into gravimetric solution, but used to independently validate the result. Authors do not offer such validation.

3/ Even if the geophysical application of this study is not substantiated, it is clear that the geodetic relevance is also not fully fulfilled. This is evident from Fig. 7, showing differences between the gravimetric and geometric (GPS/levelling) quasigeoid heights that are biased differently for each country. In gravimetic quasigeoid modelling, the final step is required to combine gravity and GPS/levelling data to remove such systematic bias. This step is missing and study is therefore not completed.

Overall, the application of multi-layer instead of single-layer approach cannot justified the publication in research-focused journals mainly due to a low scientific impact.
* * *

---

## Short Comment (SC1) · 20 Feb 2018

With great interest, I read the article of Wu et al. "A multilayer approach and its application in modeling QGNSea V1.0: a local gravmetric quasi-geoid model over the North Sea". Unfortunately, the paper lacks many details which makes it hard to assess the results. My main concern is, however, that the authors are not very consistent compared to their previous study presented in Wu et al. 2017b. In that study, they used beside shipboard and terrestrial gravity anomalies also airborne gravity disturbances, multi-satellite altimetry measurements, and GOCE gravity gradients to compute a quasi-geoid model. To validate that model, the same GPS/leveling datasets are

used as the ones used in this study. If we compare the statistics of solution A (obtained with the single-scale approach) in Wu et al. 2017b (solution computed without the use of GOCE gravity gradients) they obtained in terms of standard deviation 1.8, 1.8, and 1.6 cm for the Netherlands, Belgium, and Germany respectively. In this paper, they obtained using the single-scale approach 1.2, 2.8, and 2.9 cm for the Netherlands, Belgium, and Germany respectively. These differences are huge! Using their multi-scale approach, they obtained 0.9, 2.2, and 2.1 cm for the Netherlands, Belgium, and Germany respectively. Hence, except for the Netherland this solution still has a lower quality compared to what the authors presented in Wu et al. 2017b. The differences become even larger in case I compare their solutions obtained including GOCE gravity gradients data. To me, this shows that apparently the use of different layers of SRBFs is not the main issue in obtaining a better quasi-geoid model. Below, I provide some other concerns.

pp 2: In the first paragraph the authors state (pp 2: 4-5): "However, one layer of SRBF's parameterization may be only sensitive to parts of signals' spectrum and reduce the quality of the solution." –> This may seems so if you look to the spectrum of the SRBFs being used. However, several authors (e.g., Slobbe 2013) have successfully computed quasi-geoid solutions using one or two layer(s) of SRBFs that have an accuracy comparable or even better than the authors present in this paper. The only prerequisite is that the energy in the data at the lowest and highest frequencies is reduced by using a reference GGM and a digital terrain model, respectively. (Slobbe, D. C. (2013), Roadmap to a mutually consistent set of offshore vertical reference frames, Ph.D. thesis, Delft University of Technology.)

pp 2: I somehow have difficulties in understanding the main objective of this paper. The authors state without motivation (pp 2: 23-26): "However, differing from these methods mentioned above, we propose a multilayer approach, inspired by the power spectral analysis of local gravity observations, which indicates the gravity signals are the sum of the contributions generated from the anomaly sources that locate at different depths."

In my opinion, a proper motivation is required. It should become clear what are the limitations in existing multi-resolution representation/multi-scale approaches and how the approach proposed by the authors is going to tackle these. Definitely, the authors are not the first ones that utilize a multi-scale approach as they mention themselves.

Section 2.1: It is not entirely clear to me whether or not the authors used GOCE gravity gradients as an additional datasets as they did in Wu et al. 2017b? The confusion is introduced by their sentence (pp 19: 6-8): "Moreover, the improvements in the frequency bands that GOCE data contribute may be also the reasons, since EGM2008/EGG08 was developed without GOCE data." This suggests that they used it. However, the dataset is not mentioned in Section 2.1. And what about the radar altimeter data and airborne gravity data the authors used in Wu et al. 2017b? If, indeed, these datasets are not used. What is the reason for that? In the abstract the authors mention that "A multilayer approach is set up for local gravity field modeling based on the idea of multi-resolution representation merging heterogeneous gravity data." What they do understand by "heterogeneous"? With their approach, can they not handle different data types?

pp 6: From Figure 1, the authors conclude that "the gravity signals are the superstition (should be "superposition" I guess) of the contributions generated from the anomaly sources at different depths; and the signals originated from different anomaly sources have heterogeneous spectral contents". I have strong doubts. In Figure 1, I observe a quite smooth spectrum (no distinct peaks or whatsoever). The red lines are to me somewhat artificial.

pp 6: It is not clear how the authors estimated/obtained $A\_W$ (first term of Eq. 4)? Given Eqs. 6-7, I suppose $A\_W$ is not estimated...?

pp 8: To compute their solutions, the authors applied variance component estimation and regularization. However, nowhere the regularization parameter is given, neither the estimated weights.

pp 8: It is not clear why the authors used 10 as the "preliminary maximum order for decomposition"? Why not 20 or 5?

pp 10: In the manuscript, the authors suggest that the wavelet details (D_W) have a kind of geophysical interpretation; for example, D_W is explained as "the local anomaly originated from shallow and small-scale heterogeneous substances." If so, can the authors comment on the maps shown in Figure 2? To me, these are very peculiar. In particular D_5, D_6, and D_7 show strange stripy patterns...

pp 13: With Figure 4, I have the same problem as I have with Figure 1. How they came up with the red lines?

pp 15: The authors mention without any motivation that "Point-wise terrestrial and shipboard gravity anomalies are merged for modeling." Why, these datasets usually have different accuracies...

pp 15-16: "These results demonstrate that the multilayer approach can more accurately recovers the local high-frequency signals than the single-layer one." –> Of course, the least-squares residuals are lower! In the multilayer approach you locate the SRBFs much shallower!

Table 6: The authors have used GNSS/leveling data to validate their quasi-geoid model. What is not clear to me at all is why the statistics presented in Table 6 for the single-layer approach are so different from the values they presented in Wu 2017b (solution A). In that paper, they obtained in terms of standard deviation 1.8, 1.8, and 1.6 cm for the Netherlands, Belgium, and Germany respectively. The parametrization they have used is the same. In this paper, they obtain 1.2, 2.8, and 2.9 cm for the Netherlands, Belgium, and Germany respectively. These differences are enormous! Can the authors explain what happened? Is that due to the fact that you did not use radar altimeter and airborne gravity data, and merged shipboard and terrestrial data sets. Anyway, it seems that compared to their work presented in Wu 2017b, their multi-scale approach performs still worser (except for the Netherlands)!

pp 19: "Apart from the application of different techniques for modeling, these differences are partly interpreted as the additional signals introduced by QGNSea V1.0, stemming from the incorporation of more high-quality gravimetry". This maybe applies to EGM2008 and EIGEN-6C4, but not to EGG2008.

Figure 8, the analysis is hampered by edge effects in QGNSea V1.0. The authors should exclude the edges of the area over which they computed QGNSea V1.0.

The derived MDT models are not realistic. Please use DTU13MSS and EGG2008 to compute a MDT model and compare that to the one obtained using DTU13MSS and QGNSea V1.0. Prominent signals, like the Norwegian coastal current are not visible at all (e.g., Idžanovi Ìą̧c 2017)! (Idžanovi Ìą̧c, M., V. Ophaug, and O. B. Andersen (2017), The coastal mean dynamic topography in Norway observed by CryoSat-2 and GOCE, Geophys. Res. Lett., 44, 5609–5617, doi:10.1002/ 2017GL073777.)

---

## Referee Comment (RC2) · Anonymous Referee #2 · 10 Apr 2018

I have read the interesting manuscript "A multi-layer approach and its application in modeling QGNSea V1.0: a local gravmetric quasi-geoid model over the North Sea" by Yihao Wu, Zhicai Luo, Bo Zhong, and Chuang Xu. The manuscript focuses on a multi-layer approach compared to a single layer approach in the computation of the local gravity geoid.

I have the following comments: 1. Muliti layer approach gives (according to Table 5 and Fig 7 and page 16-17) a better fit than single layer approach. The fit would naturally increase with incrasing level of parameters, but it is statistical significant. A statistical test such as AIC (Akaike information criterion) or BIC would give valuable information.

[Figure]

2. For a better comparison with EGG08 the same or similar global geopotential model should be used.

3. Figure 2: A comment related to the different patterns observed in Figure 2 would be of interest.

---

## Author Comment (AC1) · 24 Jun 2018

Anonymous Referee #1 Authors present elegant and well-written numerical study for the SRBF gravimetric quasigeoid modelling using the multi-layer approach and compared results with a single-layer approach. This case study is very suitable for geodetic proceedings, but the modelling of quasigeoid surface is out of geophysical interest. This is main reason I recommend rejection of this article. Authors attempt to add some geophysical content (page 12/ line 14 to page 13/ line 5) is irrelevant. This is also

evident from gravity signal decomposition in Fig. 2 that does not reflect any real geological features, rather than reflects the properties of kernel for different depths. There are additional major issues to be addressed by authors before considering further publication.

Response: The authors thank the reviewer for these beneficial comments. Before discussing the geophysical meaning of this study, the authors would like to introduce its motivation. With aspect to new modeling approach development, we develop a new parameterization of SRBFs' network for regional gravity field recovery. Based on the idea multi-resolution representation, we not only parameterize the multi-scale method in a mathematical way, but also linked the detailed signals to the anomaly sources at different depths beneath the topography, which are recovered by the different layers. To our knowledge, no existing researches studied this issue. From this point, we believe this study may be within the scope of "Geoscientific Model Development", since we notice that describing developments such as new parameterizations is one of scopes of this journal, please see the information in https://www.geoscientific-model-development.net/. Besides, to our knowledge, no direct comparisons have been made between the single-layer approach and multi-scale one regarding the performances in local gravity. In this study, we assess the performances of the multilayer approach and traditionally-used single-layer one, where the advantages and disadvantages of different methods are analyzed. According to the reviewer's comments, we enhance the relevant part the updated manuscript and make the motivation more clearly, please see pp 2-3 in the revised version. While, for the geophysical meanings of this study, the authors think there may have several aspects we can contribute. First, local gravity field is helpful for many applications in geodesy and geophysics, e.g., studying the structure of lithosphere and ocean circulation, and a new parameterization of local gravity field may be beneficial for this issue, which can be used as the inputs for geophysical applications. Moreover, we also compute the mean dynamic topography based on the gravimetric quasi-geoid modeled in this study, which can be used for studying the ocean circulation and mass transport in the North Sea. We also enhance this part
based on the reviewer's comments, please refer to pp 27-29 in the updated version.

Yes, the authors believe the reviewer is right regarding this gravity signal decomposition in Figure 2 (in the original version) didn't include enough real geological features, and the statements in page 12/ line 14 to page 13/ line 5 didn't provide enough geophysical information for the patterns of these wavelet details in the original manuscript. However, the motivation of this study is to develop a new parameterization of gravity field based SRBFs in the framework of MRR, and the wavelet analysis is used to separate the contributions of different anomaly sources, which is finally used to design the parameterizations of multiply layers. And, the detailed investigation of the structure of lithosphere using the wavelet method is out the scope of this study. The author believe our work may contribute to study the geophysical features of bodies beneath the topography if we provide a better gravity field, however, this is not the main target for this study. However, according the reviewer's comments, we also provide the geophysical evidences for the demonstrated patterns of decomposed wavelet details and approximation (see Figure 1 and 2 in the updated version), and we believe these decomposed gravity anomalies can reveal the tectonic structure of study area at different depths. Please refer to the information in pp13-14 in the revised version.

1/ The values of variance factors for different types of observations are not given, so final accuracy and -most importantly - the claim that multi-layer approach provides better accuracy is not justified. This is especially evident from Table 5, where achieved accuracy in terms of gravity residuals is much too optimistic, because errors of gravity observations (especially for ship-borne data) are larger.

Response: Thanks the reviewer for the comments. Yes, we believe the reviewer is right, and the variance factors for different types of observations are important. According to the reviewer's comments, we add this information in the updated version, please see pp 17. For justifying the accuracies of different approaches, we actually consider several aspects. First, we check the data residuals after the least squares adjustment, and we agree with the reviewer's statement, we can't not confirm the multilayer approach works better even we derive a better fit of the data due to the noise level of gravity observations. Besides, since these data have been used for modeling, thus the comparison of SD values of data residuals can only be considered as the internal validation, not the external one. Thus, we introduce another high-quality independent data, i.e., GPS/leveling data, for validations in terms of quasi-geoid height. And, the associated validation results with GPS/leveling data, see Figure 6 and Table 6 in the updated version give us more confidence for the performances of different approaches. According to the reviewer's comments, we modify and enhance this part, please refer to pp 18-23 in the updated version.

2/ Another aspect related to validation of results is the ability of realistically extrapolating the gravity field. For this purpose sets of control point is chosen with given values that are not incorporated into gravimetric solution, but used to independently validate the result. Authors do not offer such validation.

Response: The authors thank the reviewer for these beneficial comments. We agree with the reviewer that the important aspect for the validation of results is extrapolating the gravity field, which is comparing the predicted values derived from the gravity model (e.g., model from the multilayer or single-layer approach) and ones derived from independent survey/measurements. For this aspect, we use independent GPS/leveling data for validating the result in terms of quasi-geoid heights, which is actually test the ability of the computed gravity field for realistically extrapolation. Let us explain it in more details, for modeling the regional gravity field using multilayer/single-layer approach, only the terrestrial and shipboard gravity data in terms of gravity anomalies are used, and no GPS/leveling data are combined. Then, after we solving the lease squares equation, i.e., eq.(8), we compute the unknown coefficients of SRBFs, and in this way, the regional gravity field model parameterized by SRBFs is known. Then, we use the independent GPS/leveling data for externally validate the regional SRBFs models. Since the GPS/leveling data are provided in terms of quasi-geoid heights, and their 3D coordinates don't coincide with the positions of gravity data, we need to
reconstruct the SRBFs model based on the computed SRBFs' coefficients and coordinates of GPS/leveling data, e.g., see eq.(6), and compute the gravimetric quasi-geoid heights, which are actually ones derived from the gravity field model. In the meanwhile, we also have the measured geometric quasi-geoid heights from the high-quality GPS survey and leveling measurements, which are the observed values. Then, we compute the standard deviation (SD) of the point-wise difference between GPS/leveling data and the gravimetric quasi-geoid height from the regional approach, which is actually external validation. We have thousands of GPS/leveling points over the target region, and these statistics support the results for validation of different regional models. According to the reviewer's comments, we enhance this part in the updated manuscript, please refer to pp 20-22 in the updated version.

3/ Even if the geophysical application of this study is not substantiated, it is clear that the geodetic relevance is also not fully fulfilled. This is evident from Fig. 7, showing differences between the gravimetric and geometric (GPS/levelling) quasigeoid heights that are biased differently for each country. In gravimetic quasigeoid modelling, the final step is required to combine gravity and GPS/levelling data to remove such systematic bias. This step is missing and study is therefore not completed.

Response: Thanks the reviewer for these beneficial comments. We agree with the reviewer's comments that there are biases between the modeled purely gravimetric quasi-geoid and local GPS/leveling data, mainly due to the commission errors in the GGM and uncorrected systematic errors in the local gravity data and leveling system. These biases also show up when we compare the local GPS/leveling data and existing gravimetric solutions (e.g., EGG08, EGM2008, and EIGEN-6C4). Generally, corrector-surface (Fotopoulos 2005; Nahavandchi and Soltanpour 2006) or more complicated algorithms, e.g., least squares collocation (Tscherning 1978) and boundary-value methodology (Klees and Prutkin 2008; Prutkin and Klees 2008), can be applied to reduce systematic errors and properly combine GPS/leveling data and gravimetric solutions. Also, the authors proposed a direct approach to properly combine GPS/leveling

data with the gravimetric quasi-geoid/geoid, where GPS/leveling data are treated as an additional observation group to form a new functional model, see Wu et al. (2017a). However, the target for this study is to develop a multilayer approach for gravimetric quasi-geoid modeling, which is served as a basic surface for geophysical applications, e.g., study the ocean circulation and structure of lithosphere. While, after implementing these methods for combining local GPS/leveling and gravimetric model, the derived quasi-geoid is not purely gravimetric, e.g., see the case in Wu et al. (2017a). Besides, we only have the well distributed GPS/leveling data in the limited region, i.e., in Netherlands, Belgium, and Germany; while, in other regions, no high-quality data are available. Thus, if we use the locally distributed GPS/leveling data for removing these systematic errors and computing the combined quasi-geoid, the final solution may be distorted in other regions, especially in the ocean parts, since no control data in these regions have been combined. And, this may be detrimental for geophysical applications in this area, e.g., investigating the ocean circulation in the North Sea. Over all, based on the reviewer's comments, we enhance the relevant part and add the necessary information, please refer to pp 21-22 in the revised version.

Overall, the application of multi-layer instead of single-layer approach cannot justified the publication in research-focused journals mainly due to a low scientific impact.

Response: Thanks for the reviewer's comments. First, we notice that the model development approach may coincide with the scope of "Geoscientific Model Development", and we also see describing developments such as new parameterizations is one of scopes of this journal. Moreover, we develop a new parameterization of SRBFs' networks for local gravity field modeling based on the idea of MRR, inspired by the power spectrum analysis of local gravity signals. Instead of constructing the multi-scale method in a purely mathematic way, we link the different detailed signals to the anomaly sources located at different depths, which are recovered by the various SRBFs' layers. To our knowledge, no existing literatures studied this issue. Besides, we directly compare the performances the multilayer approach and

single-layer one, and this may also provide references for assessing the advantages and disadvantages of different methods. In addition, for justifying the performances of different approaches, four aspects are considered in this study. First, from the spectrums of different approaches, i.e., Figure 4 in the new version (Figure 5 in the original one), we notice that the single-layer approach is only sensitive to parts of the signals' spectrum; while, for the high-frequency band, this approach is less sensitive. However, the multilayer approach effectively covers the spectrum of the local gravity signals, which is both sensitive to the low- and high-frequency bands. This gives us the original insight for the performances of different approaches from a theoretical perspective of view. Then, we check the data residuals after the least squares adjustment, which show the multilayer approach fits the data better, especially in regions with strong topography variations, where the high-frequency signals correlated with local topography dominate the small-scale features of regional gravity field. And, this result also coincides with the analysis of spectrums of different approaches, where the multilayer approach is more sensitive to the high-frequency bands. However, based on the reviewer's comments, we admit that the analysis of data residuals can't be treated as the criteria for justifying the performances of different approaches, since these gravity data have been used for modeling purpose, and the SD values for the data residuals derived from different methods should be the internal agreement. Besides, due to the limitation of the accuracies of gravity data, we can't make conclusions too firmly only depends on the analysis of data residuals. Moreover, based on the comments of Referee #2, we implement a Akaike information criterion (AIC) test for different models. AIC rewards the goodness of fit of data, but also includes a penalty with the increasing of the number of estimated parameters. In other words, it deals with the trade-off between the goodness of fit of the model and the simplicity of the model. AIC value is an estimator of the relative quality of statistical models for a given set of data, providing a means for model selection, and the model that gives the minimum AIC value may be more preferable (Akaike, 1974; Burnham and Anderson, 2002). The associated results demonstrate that the multilayer model gives

a smaller AIC value, which reaches a better balance between the goodness of fit of data and the simplicity of the model. This gives us the value information regarding the performances of different approaches in the view of statistical test, please see pp 19 for details in the revised manuscript. In addition, we test the test the ability of realistic extrapolation of different regional models recovered from various methods, where another independent data set, i.e., GPS/leveling measurements, is introduced for external validation. From these results, we see that the multilayer approach not only lead to a reduction for the data residuals in the least squares adjustment, but also derives a better solution assessed by the independent control data, compared to the single-layer approach. Based on these results, the authors believe this study may contribute to the literatures. Based on the reviewer's comments, we restructure the relevant parts and add the necessary information, please refer to the revised version.

Please also note the supplement to this comment:
https://www.geosci-model-dev-discuss.net/gmd-2017-289/gmd-2017-289-AC1-supplement.zip

---

## Author Comment (AC2) · 24 Jun 2018

I have read the interesting manuscript "A multi-layer approach and its application in modeling QGNSea V1.0: a local gravmetric quasi-geoid model over the North Sea" by Yihao Wu, Zhicai Luo, Bo Zhong, and Chuang Xu. The manuscript focuses on a multi-layer approach compared to a single layer approach in the computation of the local gravity geoid.

I have the following comments: 1. Muliti layer approach gives (according to Table 5 and Fig 7 and page 16-17) a better fit than single layer approach. The fit would naturally increase with incrasing level of parameters, but it is statistical significant. A statistical test such as AIC (Akaike information criterion) or BIC would give valuable information.

Response: The authors thank the reviewer for this beneficial comment. Yes, the authors totally agree with the reviewer's comment, and the fit with the data using the multilayer approach with more parameters naturally increase from the view of statistical analysis. We believe it is a very good suggestion for implementing the Akaike information criterion (AIC) or Bayesian information criterion (BIC) test of different models. In this study, we implement the AIC test, which may provide value information for model selection in another aspect. AIC rewards the goodness of fit of data, but also includes a penalty that is an increasing function of the number of estimated parameters. It deals with the trade-off between the goodness of fit of the model and the simplicity of the model. AIC test is an estimator of the relative quality of statistical models for a given set of data, providing a means for model selection, and the model that gives the minimum AIC value may be more preferable (Akaike, 1974). The AIC value of the model is defined as $AIC=2k-2\ln(L)$, where $k$ is the number of estimated parameters in the model, and $L$ is the maximum value of the likelihood function for the model (Burnham and Anderson, 2002). For gravity field modeling in this study, we work within the framework of least squares adjustment, i.e., the unknown coefficients of Poisson wavelets of different approaches (the multilayer and single-layer approach) are computed through the least squares method. We also assume that the data residuals derived from different approaches are distributed according to independent identical normal distributions with zero mean values, also see the information of data residuals in Table 5 in the revised manuscript. Then, the maximum likelihood estimate for the variance of a model's residuals distributions is RSS/n, where RSS is the residual sum of squares (RSS), and $n$ is the number of observations (Burnham and Anderson, 2002). Then, the AIC value of model is given as $AIC=2k+n\ln(RSS/n)+C$, and $C$ is a constant independent of the model (Burnham and Anderson, 2002). Since only differences in AIC are meaningful, the constant C can be ignored, and we can conveniently take AIC=2k+nln(RSS/n) for model comparisons. In this study, we compare the performances of the multilayer and single-layer model through the AIC test. In details, the number of gravity observations is 894649, and the numbers of estimated parameters in the multilayer and single-layer model are 47504 and 19477, respectively. The RSS values for the multilayer and single-layer model are $8.8527 \times 10^5$ mGal$^2$ and $1.3296 \times 10^6$ mGal$^2$, respectively, based on the data residuals after the least squares adjustment. Then, the AIC values for the multilayer and single-layer model are estimated as 85581 and 393400, respectively. Based on these statistics, we notice that the multilayer model gives a smaller AIC value, which may be more preferable since it reaches a better balance between the goodness of fit of data and the simplicity of the model. According to the reviewer's comments, we add the information of AIC test in the revised manuscript, please refer to the abstract (pp 1) section 3.3 (pp 19), conclusion (pp 30), and the Appendix (pp 32) in the updated version.

2. For a better comparison with EGG08 the same or similar global geopotential model should be used.

Response: The authors thank the reviewer for this beneficial comment. For further validate the quality of QGNSea V1.0, we compare it with other existing models, where a regional model call EGG08 and other global geopotential models (GGMs) are introduced. EGG08 is a regional gravimetric quasi-geoid model covers most areas in Europe; this model was recovered by stokes integral based on locally distributed gravity data, which was provided in terms of gridded data instead of spherical harmonics like GGMs (e.g., EGM2008 and EIGEN-6C4), and the space resolution of which is 1 minute in latitude and 1.5 minute in longitude, see Denker (2013). We also use other global geopotential models for comparisons since the authors don't have access to other regional gravimetric quasi-geoid models; for example, a new Europe gravimetric quasi-geoid called EGG2015 has been implemented (Denker, 2015), however, this model is seems not publicly available. Thus, the two high-order GGMs, i.e., EGM2008 (d/o 2190) with

the spatial resolution of 5 minute by 5 minute, EIGEN-6C4 (d/o 2190) with the spatial resolution of 5 minute by 5 minute are incorporated for further comparisons, since these two models have relatively higher spatial resolutions and better accuracies compared to most of other available GGMs, when compared with the globally distributed GPS/leveling data, see the information in http://icgem.gfz-potsdam.de/home. However, according to the reviewer's comments, we introduce another two recently published high-order GGMs (i.e., GECO (d/o 2190) (Gilardoni et al. 2015), and SGG-UGM-1 (d/o 2159) (Liang et al. 2018)), which were developed by combining GOCE data into EGM2008, for further comparisons. We also restructure and modify the relevant parts in the updated manuscript based on the reviewer's comments, please see pp. 24-27 in the revised version.

3. Figure 2: A comment related to the different patterns observed in Figure 2 would be of interest.

Response: The authors thank the reviewer for the comment. First of all, the authors believe the original wavelet details with stripe like patterns shown in Figure 2 are problematic (also see the interactive comments from the third referee), since we carefully check the source code for wavelet decomposition, and find bugs that may derive incorrect wavelet details. Based on the reviewer's comments, we redo the wavelet decomposition after the removal of bugs of source code, and compute the new wavelet details and approximation, please refer Figure 1 in the updated version, i.e., in pp 11, where no strange stripy patterns occur. Moreover, we provide the geophysical evidences for the patterns of different wavelet details. More specifically, $D\_1$ and $D\_2$ and are seems dominated by the high-frequency signals correlate strongly with the local topography, which are mainly due to the uncorrected topographical signals in RTM corrections. $D\_3$ and $D\_4$ with respective average source depths of 4.5 km and 9.2 km primarily reflect the density distribution of the upper crust. The distribution of $D\_5$ and $D\_6$ is in agreement with the tectonic structure of the middle crust. $D\_7$ is consistent with the Moho undulation. $D\_8$ and $A\_8$ represent density distribution of the

upper mantle. Overall, these decomposed gravity anomalies can reveal the tectonic structure of study area at different depths. Based on the reviewer's comments, we add the detailed comments related to the different patterns of wavelet details in Figure 1 (Figure 2 in the original version) in the revised manuscript, please see the information in pp13-14. Moreover, we notice that the wavelet details and approximation change after we implement the wavelet decomposition with the errors corrected source code, and we redo the whole procedure for the multiply layers' network design, i.e., estimating the depths of different layers and the number of Poisson wavelets in each layer. Then, we recompute the solution based on the multilayer approach with the updated parameters (i.e., the depths of different layers and the number of Poisson wavelets in each layer), and redo the comparisons with existing models based on the updated solution. Following, the geodetic MDT (called MDTNS_QGNSea) based on the updated model derived from the multilayer approach is computed. Please refer to pp 13-29 in the revised manuscript.

Please also note the supplement to this comment:
https://www.geosci-model-dev-discuss.net/gmd-2017-289/gmd-2017-289-AC2-supplement.zip

---

## Author Comment (AC3) · 24 Jun 2018

With great interest, I read the article of Wu et al. "A multilayer approach and its application in modeling QGNSea V1.0: a local gravmetric quasi-geoid model over the North Sea". Unfortunately, the paper lacks many details which makes it hard to assess the results. My main concern is, however, that the authors are not very consistent compared to their previous study presented in Wu et al. 2017b. In that study, they used beside shipboard and terrestrial gravity anomalies also airborne gravity disturbances, multi-satellite altimetry measurements, and GOCE gravity gradients to compute a quasi-geoid model. To validate that model, the same GPS/leveling datasets are used as the ones used in this study. If we compare the statistics of solution A (obtained with the single-scale approach) in Wu et al. 2017b (solution computed without the use of GOCE gravity gradients) they obtained in terms of standard deviation 1.8, 1.8, and 1.6 cm for the Netherlands, Belgium, and Germany respectively. In this paper, they obtained using the single-scale approach 1.2, 2.8, and 2.9 cm for the Netherlands, Belgium, and Germany respectively. These differences are huge! Using their multi-scale approach, they obtained 0.9, 2.2, and 2.1 cm for the Netherlands, Belgium, and Germany respectively. Hence, except for the Netherland this solution still has a lower quality compared to what the authors presented in Wu et al. 2017b. The differences become even larger in case I compare their solutions obtained including GOCE gravity gradients data. To me, this shows that apparently the use of different layers of SRBFs is not the main issue in obtaining a better quasi-geoid model. Below, I provide some other concerns.

Response: The authors thank the reviewer for these beneficial comments. To our knowledge, the solutions in this study are indeed inconsistent with ones shown in Wu et al. (2017b), and should not be made simply comparison with each other. There are several reasons that you find the accuracy of solution modeled with the single-layer approach in this study is different from the one displayed in Wu et al. (2017b). First, in this study we only use terrestrial and shipboard gravity data, no airborne or radar altimetry data are incorporated. While, for the solution A (without GOCE data) in Wu et al. (2017b), we used terrestrial, shipboard, and airborne gravity data, and radar altimetry data. Thus, even we use the same GPS/leveling data for validation, we observe the different statistics for accuracy assessment. Second, the target area in this study and the one in Wu et al. (2017b) are not consistent. The area in the study of Wu et al. (2017b) extends from 49.5°N to 56°N latitude and 0.25°E to 8.25°E longitude (see page 6 in Wu et al., 2017b); While, in this study we choose a much larger area, which covers an area of 49°N-61°N latitude and -6°E-10°E (see page 3 in the original

manuscript). And, when we choose a larger region, more data in UK, Norway, and the North Sea are incorporated. However, we notice that the data in Norway are sparsely distributed, especially in the mountainous regions; and this situation also occurs in the north parts of the North Sea, see Fig.2 in Wu et al. (2017b). Consequently, the quality of the solution may be affected if different gravity data are introduced, even when we validate the solution only use the GPS/leveling data in the Netherlands, Belgium, and Germany. We should not directly compare these statistics if these solutions are modeled under different conditions. For the similar reasons, we can't simply compare the solutions computed in this study with the ones in Wu et al. (2017b).

pp 2: In the first paragraph the authors state (pp 2: 4-5): "However, one layer of SRBF's parameterization may be only sensitive to parts of signals' spectrum and reduce the quality of the solution." –> This may seems so if you look to the spectrum of the SRBFs being used. However, several authors (e.g., Slobbe 2013) have successfully computed quasi-geoid solutions using one or two layer(s) of SRBFs that have an accuracy comparable or even better than the authors present in this paper. The only prerequisite is that the energy in the data at the lowest and highest frequencies is reduced by using a reference GGM and a digital terrain model, respectively. (Slobbe, D. C. (2013), Roadmap to a mutually consistent set of offshore vertical reference frames, Ph.D. thesis, Delft University of Technology.).

Response: The authors thank the reviewer for these comments. Yes, we believe the reviewer's statement is right regarding this multilayers approach may work fine when only the residual gravity field is modeled from the ground-based data, i.e., the long- and short-wavelength parts have been removed. In this study, we also model the regional gravity field within the framework of remove-compute-restore method, and only the residual signals are parameterized, we emphasize this in the revised manuscript according to the reviewer's comments, see pp 2 in the updated manuscript. We also see the (one) two layers of SRBFs works fine, i.e., see Slobbe (2013) and Wittwer (2009). However, we should not compare the accuracies of the solutions if they are modeled

under different solutions, see our detailed response to Q1. We also cite the contributions of the existing literatures regarding the modeling with single-layer approach, i.e., Wittwer (2009), Slobbe (2013). Moreover, we remove the "However, one layer of SRBF's parameterization may be only sensitive to parts of signals' spectrum and reduce the quality of the solution.", since we believe this is too absolute to some extent, which may lead to the wrong understanding. Based on the reviewer's comments, we modify and restructure the relevant contents, please see pp 2 in the updated version.

pp 2: I somehow have difficulties in understanding the main objective of this paper. The authors state without motivation (pp 2: 23-26): "However, differing from these methods mentioned above, we propose a multilayer approach, inspired by the power spectral analysis of local gravity observations, which indicates the gravity signals are the sum of the contributions generated from the anomaly sources that locate at different depths." In my opinion, a proper motivation is required. It should become clear what are the limitations in existing multi-resolution representation/multi-scale approaches and how the approach proposed by the authors is going to tackle these. Definitely, the authors are not the first ones that utilize a multi-scale approach as they mention themselves.

Response: The authors thank the reviewer for these beneficial comments. Yes, we think the reviewer's comments are right. In our opinion, there are two limitations for the existing studies. First, to our knowledge, no direct comparisons have been made between the single-layer approach and multi-scale one regarding the performances in local gravity field recovery. Besides, the existing multi-scale methods mainly construct the multi-scale framework in a mathematical way, where no explicit geophysical meanings are investigated. Thus, the main contributions of this study are twofold. First, to develop a new parameterization of SRBFs network in the framework of the MRR idea, i.e., the so-called multilayer approach; and the multiply layers are linked to the anomaly sources at different depths beneath the topography, which aim at recovering the signals at different levels. To our knowledge, no existing literatures studied this issue. Moreover, we assess the performances of the multilayer approach and traditionally-used

single-layer one in this study, where the advantages and disadvantages of different methods are analyzed. According to the reviewer's comments, we modify the relevant part the updated manuscript and make the motivation more clearly, please see pp 2-3 in the revised version.

Section 2.1: It is not entirely clear to me whether or not the authors used GOCE gravity gradients as an additional datasets as they did in Wu et al. 2017b? The confusion is introduced by their sentence (pp 19: 6-8): "Moreover, the improvements in the frequency bands that GOCE data contribute may be also the reasons, since EGM2008/EGG08 was developed without GOCE data." This suggests that they used it. However, the dataset is not mentioned in Section 2.1. And what about the radar altimeter data and airborne gravity data the authors used in Wu et al. 2017b? If, indeed, these datasets are not used. What is the reason for that? In the abstract the authors mention that "A multilayer approach is set up for local gravity field modeling based on the idea of multi-resolution representation merging heterogeneous gravity data." What they do understand by "heterogeneous"? With their approach, can they not handle different data types?

Response: The authors thank the reviewer for these beneficial comments. We didn't directly use the along-track GOCE gradients as the additional groups as we did in Wu et al. (2017b). In fact, only the terrestrial and shipboard gravity data are introduced as the observation groups, Section 2.1 give the details regarding the data sets we use here. Although we didn't directly GOCE gradients, we used the GOCO05S as the reference model, which was computed with GOCE data. However, for the development of EGM2008/EGG08, no GOCE data were used. Thus, in the bandwidth that GOCE data contribute, i.e., in frequencies from 0.005 to 0.1 Hz, we believe our model may outperform EGM2008/EGG08. In this sense, we say "Moreover, the improvements in the frequency bands that GOCE data contribute may be also the reasons, since EGM2008/EGG08 was developed without GOCE data.", it doesn't not mean we directly combine the GOCE data as additional observation groups for modeling, but just use

a more accurate reference model in the measurement bandwidth (MBW) of GOCE mission. The motivation of this study is to develop a new parameterization of SRBFs network in the framework of the MRR idea, i.e., the so-called multilayer approach, and compare it with the traditionally-used single-layer approach for the performances in regional gravity field recovery. For a case study, we only use the terrestrial and shipboard gravity data, and the results in case derive reasonable solutions, which can be used for supporting the conclusions of this study. The "heterogeneous" here not only means the different types of observations, but also refer to the data sets with different spatial resolutions/coverage, different noise levels, see Wu et al. (2017c) in the updated version regarding the details of heterogeneous data sets. The different types of observations groups can be combined through the multilayer approach just similar as the way the researchers did for in the single-layer approach, e.g., see Klees et al. (2008), and Slobbe (2013).

pp 6: From Figure 1, the authors conclude that "the gravity signals are the superstition (should be "superposition" I guess) of the contributions generated from the anomaly sources at different depths; and the signals originated from different anomaly sources have heterogeneous spectral contents". I have strong doubts. In Figure 1, I observe a quite smooth spectrum (no distinct peaks or whatsoever). The red lines are to me somewhat artificial.

Response: The authors thank the reviewer for this comment. First, we only model the residual gravity signals in this study, and the power spectrum showed in Figure 1 is based on the residual gravity data in Sect 2.1, the short- and long-wavelength signals are removed. Moreover, the local gravity signals are the sum of the contributions of different anomaly sources, i.e., the contributions from different anomaly sources have been separated, and the spectrum here shows the one for the mixed signals. After we separate the different signals with wavelet decomposition, and more distinguished spectrums occur, see Figure 3 in the revised manuscript. We also want to mention that Figure 1 is just an example support the statement that the gravity signals are the sum

of the contributions of different sources, and red lines are also the illustrations show that slopes of the spectrum are different in different frequency bands, and please see our response to the question below regarding how we estimate the slopes (i.e., the red lines) of the spectrum. However, we also think this figure is confusing to some extent, and we remove this figure and restructure the relevant part based on the reviewer's comments, please see pp 6 in the updated version.

pp 6: It is not clear how the authors estimated/obtained $A\_W$ (first term of Eq. 4)? Given Eqs. 6-7, I suppose $A\_W$ is not estimated...?

Response: The authors thank the reviewer for this comment. Based on Eq.4, the gravity anomaly can be decomposed into a number of wavelet details and a wavelet approximation. Thus, the difference between the gravity anomaly and the sum of wavelet details is the wavelet approximation $A\_W$, similar information can be found in Xu et al. (2017, 2018). The target for the wavelet decomposition is to design the parameterizations of multilayer approach, and for modeling purpose, the point-wise gravity data are combined just as we do in the single-layer approach.

pp 8: To compute their solutions, the authors applied variance component estimation and regularization. However, nowhere the regularization parameter is given, neither the estimated weights.

Response: The authors thank the reviewer for this beneficial comment. Yes, we believe the reviewer is right, and the variance factors for different types of observations are important, indicate their relative contributions, and play a key role in data combination. According to the reviewer's comments, we add the information of estimated variance factors of different observations groups and regularization parameter in the updated version, please see pp 17.

pp 8: It is not clear why the authors used 10 as the "preliminary maximum order for decomposition"? Why not 20 or 5?

[Figure]

Response: The authors thank the reviewer for this beneficial comment. This is a good question. To some extent, the original maximum order is arbitrarily chosen. However, wavelet analysis has a number of nice properties, for instance, the low-order details are invariant with the increase of decomposition order, and only the high-order details and wavelet approximation change. Thus, we can preliminarily choose a predefined order for decomposition, and analyze the derived details as we do in Section 3.1. If there are still details that are useful for constructing the multilayer model haven't been separated, we need to increase the decomposition order until all the useful details have been extracted; otherwise, we can truncated to a specific order as we do in this study, and compute the corresponding the necessary details and approximation for constructing the multiply layer's network. According to the reviewer's comment, we add and enhance this information in the updated version, please see pp 9.

pp 10: In the manuscript, the authors suggest that the wavelet details ($D\_W$) have a kind of geophysical interpretation; for example $D\_W$ is explained as "the local anomaly originated from shallow and small-scale heterogeneous substances." If so, can the authors comment on the maps shown in Figure 2? To me, these are very peculiar. In particular $D\_5$, $D\_6$, and $D\_7$ show strange stripy patterns...

Response: The authors thank the reviewer for these beneficial comments. We think the reviewer's concern is right regarding these strange stripe like signals, since we carefully check the source code for wavelet decomposition, and find bugs that may derive incorrect wavelet details. Based on the reviewer's comments, we redo the wavelet decomposition based on errors corrected source code, and compute the updated wavelet details and approximation, please refer to Figure 1 (in pp 11) in the updated version, and no strange stripy patterns occur. Moreover, we provide the geophysical evidences for the different patterns of various wavelet details. More specifically, $D\_1$ and $D\_2$ are seems dominated by the high-frequency signals correlate strongly with the local topography, which are mainly due to the uncorrected topographical signals in RTM corrections. $D\_3$ and $D\_4$ with respective average source depths 4.5 km and 9.2 km

primarily reflect the density distribution of the upper crust. The distribution of D_5 and D_6 is in agreement with the tectonic structure of the middle crust. D_7 is consistent with the Moho undulation. D_8 and A_8 represent density distribution of the upper mantle. Overall, these decomposed gravity anomalies can reveal the tectonic structure of study area at different depths. Based on the reviewer's comments, we add the detailed comments related to the different patterns of Figure 1 in the revised manuscript, please see the information in pp13-14. We also notice that the wavelet details and approximation change after we implement the wavelet decomposition with the errors corrected source code, and we redo the whole procedure for the multiply layers' network design, i.e., estimating the depths of different layers and the number of Poisson wavelets in each layer. Then, we recompute the solution based on the multilayer approach with the updated parameters of multiply layers (i.e., the depths of different layers and the number of Poisson wavelets in each layer), and redo the comparisons with existing models based on the updated solution. Following, the geodetic MDT (called MDTNS_QGNSea) based on the updated model derived from the multilayer approach is computed. Please refer to pp 13-29 in the revised manuscript.

pp 13: With Figure 4, I have the same problem as I have with Figure 1. How they came up with the red lines?

Response: The authors thank the reviewer for this comment. The average depths for the power spectrum of wavelet details are estimated from the eq.(5). Actually, a number of literatures showed how to estimate the depths from these spectrums, e.g., see Figure 4 in Xu et al. (2018). More specifically, the red lines represent rates of change for logarithmic power relative to wave number, which are estimated by autoregressive method. The starting point and terminal point of the red lines are inflection points of the curves (green lines in Figure 3), recognized by us according to the trend of the curves. Based on the reviewer's comment, we also add this information in the revised manuscript, see pp 13 in the updated version.

pp 15: The authors mention without any motivation that "Point-wise terrestrial and

shipboard gravity anomalies are merged for modeling." Why, these datasets usually have different accuracies...

Response: The authors thank the reviewer for this comment. For modeling purpose, point-wise terrestrial and shipboard data are combined. These data have different accuracies, and this is also one of the reasons why we need the MCVCE method for estimating the variance factors for different observation groups. The gridded gravity data is only used for wavelet decomposition, i.e., for designing the multiply layers' network, since this wavelet decomposition method needs the regularly distributed data. While, for modeling purpose, the point-wise data are directly used just the same as the single-layer approach. We also enhance this part for avoid confusing based on the reviewer's comment, please refer to pp 17 in the updated version.

pp 15-16: "These results demonstrate that the multilayer approach can more accurately recovers the local high-frequency signals than the single-layer one." –> Of course, the least-squares residuals are lower! In the multilayer approach you locate the SRBFs much shallower!

Response: The authors thank the reviewer for this enlightening comment. Yes, we believe the lower residuals may be attributed to the shallower SRBFs. Shallower SRBFs are more sensitive to the local high-frequency signals, and the corresponding spectrum also shifts to high-frequency bands, which may lead to a better fit to the data. However, there are still two aspects may be of concern. First, we parameterized the local gravity field by 7 layers with different depths, where the layer7 are still deeper than 40 km (where we locate the single-layer of SRBFs' grid), see Table 3 in the revised manuscript, thus not all the layers are shallower than 40 km. In addition, to our experience with the single-layer approach, the shallower SRBFs' grid may lead to a reduction of least square residuals, but not guarantee a better solution, i.e., the better fit to the independent control data for external validation, please refer to Figure 2, 3 in Wu et al. (2016), which clearly shows a shallower grid than 40 km may not derive a better solution. However, in this study, the multilayer approach not only derives a

[Figure]

better fit to the data, but also obtains better solution validated by the control data. This can't acquire by solely putting the SRBFs' grid shallower. According to the reviewer's comments, and we restructure and enhance the relevant parts in the updated version, please refer to pp 18-19.

Table 6: The authors have used GNSS/leveling data to validate their quasi-geoid model. What is not clear to me at all is why the statistics presented in Table 6 for the single-layer approach are so different from the values they presented in Wu 2017b (solution A). In that paper, they obtained in terms of standard deviation 1.8, 1.8, and 1.6 cm for the Netherlands, Belgium, and Germany respectively. The parametrization they have used is the same. In this paper, they obtain 1.2, 2.8, and 2.9 cm for the Netherlands, Belgium, and Germany respectively. These differences are enormous! Can the authors explain what happened? Is that due to the fact that you did not use radar altimeter and airborne gravity data, and merged shipboard and terrestrial data sets. Anyway, it seems that compared to their work presented in Wu 2017b, their multi-scale approach performs still worser (except for the Netherlands)!

Response: The authors thank the reviewer for this comment. In our opinion, we should not directly compare these statistics if these solutions are modeled under different conditions. The solution derived from single-layer/multi-layer approach should be different from the solution A in Wu et al. (2017b), since the inputs for these solutions are inconsistent. Thus, even we use the same GPS/leveling data for validation, the derived statistics are heterogeneous. Please see our detailed response to the first question.

pp 19: "Apart from the application of different techniques for modeling, these differences are partly interpreted as the additional signals introduced by QGNSea V1.0, stemming from the incorporation of more high-quality gravimetry". This maybe applies to EGM2008 and EIGEN-6C4, but not to EGG2008.

Response: The authors thank the reviewer for the comment. Yes, we believe the reviewer is right. We also refer to EGM2008/EIGEN-6C4 when we say the additional

signals introduced by QGNSea V1.0 are stemmed from the incorporation of more high-quality gravimetry. And, the sentence "Apart from the application of different techniques for modeling, these differences are partly interpreted as the additional signals introduced by QGNSea V1.0, stemming from the incorporation of more high-quality gravimetry" further explains "For EGM2008/EIGEN-6C4, remarkable differences show in south of Norway and northwest of Germany". However, according to the reviewer's comments, we modify this part slightly to eliminate misunderstanding, see pp 24 in the updated version.

Figure 8, the analysis is hampered by edge effects in QGNSea V1.0. The authors should exclude the edges of the area over which they computed QGNSea V1.0.

Response: The authors thank the reviewer for this comment. Yes, we believe the reviewer is right that the edge effects should be excluded. In fact, for plotting Figure 8 in the original manuscript (Figure 7 in the revised version), we have excluded the edge effects by contracted by 0.5 degree in all the directions. For modeling purpose, the boundary limits for the target area is chosen as 49 N-61 N latitude and -6 E-10 E longitude, see sect 2.1. While, for displaying the differences between different models, the signals only inside 49.5 N-60.5 N latitude and -5.5 E-9.5 E longitude have been extracted and compared. We also add this information in the updated version, please refer to pp 24.

The derived MDT models are not realistic. Please use DTU13MSS and EGG2008 to compute a MDT model and compare that to the one obtained using DTU13MSS and QGNSea V1.0. Prominent signals, like the Norwegian coastal current are not visible at all (e.g., Idžanovi ÌAËŽc 2017)! (Idžanovi ÌAËŽc, M., V. Ophaug, and O. B. Andersen (2017), The coastal mean dynamic topography in Norway observed by CryoSat-2 and GOCE, Geophys. Res. Lett., 44, 5609–5617, doi:10.1002/ 2017GL073777.)

Response: The authors thank the reviewer for these beneficial comments. Yes, we agree with the reviewer's comments, and the geodetic MDTs in the original manuscript

are not realistic. The problem is seems due to the implementation of too strong filtering on the raw MDTs. In the original manuscript, we compared the MDT derived from QGNSea V1.0 with the existing global model called DTU13MDT. DTU13MDT was computed in a purely geodetic way, where the difference between DTU13MSS and the quasi-geoid derived from EGM2008 was used to estimate the raw MDT, and the derived MDT was further smoothed by a Guassian filter with a correlation length of 75 km to suppress the small-scale signals (Andersen et al., 2013). To make these comparisons consistently, in the original manuscript, the computed raw MDT (the difference between the DTUMSS13 and QGNSea V1.0) was also filtered by a Guassian filter with a correlation length of 75 km. However, based on the reviewer's comments, we believe this filter may be too strong since the prominent signals have been filtered out. According to the reviewer's comments, in the revised manuscript, we compute the raw MDT by computing the difference between DTUMSS13 and QGNSea V1.0/EGG08, and filter the raw MDT by a Gaussian filter to further smooth the derived MDT, which is called as MDTNS_QGNSea/MDTNS_EGG08. Considering the small-scale signals that have the wavelengths shorter than several kilometers can't be recovered from the local gravity data, since the mean distance between gravity data is approximately at 6∼7 km level, the correlation length of Gaussian filter is chosen as 6 km instead of 75 km in the revised manuscript. This time, the derived MDTs show more realistic patterns, although MDTNS_QGNSea don't provide a full picture of Norwegian coastal currents due to the limited data coverage in Norway and its neighbouring ocean areas, please see Figure 9 in the updated version. According to the reviewer's comments, we restructure and modify the part for MDT comparison, please refer to pp 27-29 in the new version.

Please also note the supplement to this comment:
https://www.geosci-model-dev-discuss.net/gmd-2017-289/gmd-2017-289-AC3-supplement.zip

---

## Author Response (AR2)

**Response to the Referees' Comments**

First of all, we would like to give our sincerest thanks to the reviewers for the beneficial suggestions and comments, and we deeply appreciate your contributions, which help us for correcting and improving the manuscript. Our responses are listed as follows by using the red fonts. If there are still unclear or incorrect parts, the authors are very willing to make further corrections and improvements based on the reviewer's comments. Thanks again for your contributions.

Comment on "A multilayer approach and its application in modeling QGNSea V1.0: a local gravmetric quasi-geoid model over the North Sea" by Yihao Wu et al.
Anonymous Referee #3
Authors used the multilayer approach to model QGNSea V1.0 based on SRBF from the local terrestrial and ship-borne gravimetric data in the manuscript. The English writing should be further improved. The results may be of geodetic science sense by comparing with the local gravity data, GPS/leveling data, and the Earth gravity field models.

Response: The authors thank the reviewer for these beneficial comments. Based on the reviewer's comments, the authors have asked a native speaker to make a thorough language check of this manuscript, and we corrected all the grammar errors and bad language usages to improve its English level, please refer to the revised manuscript.

[1] Section 2.1 Study area and data. Here the local terrestrial and ship-borne gravity data are used in the study. Why not use the airborne gravity data and the satellite altimeter data? These data ever were used in Wu et al. 2017. Dr Wu is the first author in the manuscript. The results in the manuscript may be worse than those in Wu et al. 2017. More data can help to improve the resolution and the precision of QGNSea model. How to unify the datum of all data in the study?

Response: The authors thank the reviewer for these beneficial comments. Yes, we think the reviewer is right that the incorporation of more data help improve the resolution and the precision of QGNSea model. The solutions modeled in this study are inconsistent with ones shown in Wu et al. (2017c), since the input data and study area are different in these two studies. Yes, the solution modeled in this study may show worse results than the one displayed in Wu et al. (2017c), e.g., see the validation results in Belgium and Germany. The reasons are twofold. First, in this study we only use terrestrial and shipboard gravity data, no airborne or radar altimetry data are incorporated. While, for the solution A (without GOCE data) in Wu et al. (2017c), we used terrestrial, shipboard, and airborne gravity data, and radar altimetry data. Second, the target area in this study and the one in Wu et al. (2017b) are not consistent. The area in the study of Wu et al. (2017c) extends from 49.5 °N to 56 °N latitude and 0.25 °E to 8.25 °E longitude (see page 6 in Wu et al., 2017c); While, in this study we

choose a much larger area, which covers an area of 49°N-61°N latitude and -6°E-10°E (see page 3 in the original manuscript). And, when we choose a larger region, more data in UK, Norway, and the North Sea are incorporated. However, we notice that the data in Norway are sparse, especially in the mountainous regions, and this situation also occurs in the north parts of the North Sea, see Fig.2 in Wu et al. (2017c). Consequently, the quality of the solution may be affected if different gravity data are introduced, even when we validate the solution only use the GPS/leveling data in the Netherlands, Belgium, and Germany.

The motivation of this study is to develop a new parameterization of SRBFs network in the framework of the MRR idea, i.e., the so-called multilayer approach, and compare it with the traditionally-used single-layer approach for the performances in regional gravity field recovery. For a case study, we only use the terrestrial and shipboard gravity data, and the derived results show reasonable solutions, which can be used for supporting the conclusions of this study. For the time beings, the authors put effects on deriving reasonable solutions by combining heterogeneous gravimetry and altimetry data as well as GOCE data through the multilayer approach, and compare these solutions with ones modeled from the single-layer approach. However, more open issues need to be investigated since GOCE data mainly contribute to low-frequency bands of gravity field, and deeper layers than ones we use to combine surface data may be implemented to incorporate these satellite observations. The initial solutions seem to be promising, hopefully, we can obtain reasonable results, and make a detailed discussion regarding these issues in another paper. According to the reviewer's comments, we introduce this information in the conclusion part, see pp 32-33 in the revised version.

Yes, we need to unify the horizontal and vertical datums for all the gravity data, since these data were derived from various institutions over different time spans (see Wu et al. (2017c)). The European Terrestrial Reference System 1989 (ETRS89) and European Vertical Reference Frame 2007 (EVRF2007) are chosen as the horizontal and vertical systems (Slobbe, 2013), respectively. For terrestrial data in all regions except for the UK, transformation parameters can be directly used to unify the national vertical systems to EVRF2007. For shipborne data, the reference models derived from DTU13, i.e., mean sea surface (MSS) and mean dynamic topography (MDT), are used to extract the ellipsoidal and orthometric/normal height information. With respect to terrestrial data in the UK, the effects of the systematic errors are in its leveling system, where both the south-north slope and regional distortions exist, and no valid transformation parameters can be used for vertical datum unification (Penna et al., 2013). However, as EGM2008 has no slope when compared with the local corrected mean sea level values (Penna et al., 2013), we use it as the reference model for data reduction. Details can be found in Wu et al. (2017c). According to the reviewer's comments, we add the brief information for horizontal and vertical datums unification in the revised version, please see pp 4.

[2] Section 3 Numerical results and discussion. The full flow chart to build QGNSea model should be shown in detail, which can make readers to understand the technological idea in general.

Response: Thanks the reviewer for the comments. Yes, it is a very nice suggestion to build a full flow chart for model development based on the multilayer approach, which is pretty beneficial for potential readers. According to the reviewer's comment, we provide a detailed flowchart of designing the multilayer model, see Figure 1 in pp 11 in the revised manuscript. Also, a brief description is added, see pp 10 in the updated version.

[3] Section 3.1. What algorithm can be used to estimate automatically the order of wavelet analysis? What criterion can be used to judge the optimal order?

Response: The authors thank the reviewer for these beneficial comments. This is a very good question regarding the determination of optimal decomposition order. The authors believe it is still an open issue regarding the automatic determination of the optimal order for wavelet analysis. Usually, we preselect an order for wavelet decomposition, and this order is arbitrarily chosen to some extent. Then, we need to analyze the decomposed wavelet details and approximation for determining the optimal order. If there are still details that are needed for constructing the multilayer model haven't been separated, we need to change the decomposition order until all the useful details have been extracted; otherwise, we truncate to a specific order, and compute the wavelet details and approximation to construct the multiply layer's parameterization. Please see the details in Section 3.1 about how to choose the decomposition order in the case of this study. Since the analysis of the wavelet signals need the background knowledge of the local gravity field signals (i.e., the spectral contents of residual gravity signals), also human interventions are necessary for key parameters estimation and crucial choices during these procedures (e.g., the types of basis functions for wavelet decomposition and number of layers), thus it is difficult for make these procedures totally automatic. The future work may involve in developing a data-adaptive algorithm, however, additional efforts are needed. According to the reviewer's comment, we enhance this information in the conclusion part, please see pp 32 in the revised manuscript.

[4] Figure 1. What geological structure and geophysical mechanism are corresponding to each layer?

Response: Thanks the reviewer for these comments. Gravity anomaly primarily reflects the density heterogeneity of anomalous mass in the Earth interior. Density distributions at different depths are strongly correlated with the geological structure. Therefore, the decomposed gravity anomalies in the Figure 2 (Figure 1 in the original version) can reveal the tectonic structure of study area at various depths.

The source depths of D1 and D2 in Figure 2 are less than 3 km. They highly correlate with the local topography, mainly due to the uncorrected topographical signals. The anomalies in the ocean are smooth, while those on land are more dispersed. D3 and D4 with the respective source depths of 4.5 km and 9.2 km are corresponding to the tectonic structure in the upper crust. The Viking Graben located in the northern of the North Sea is in agreement with the dispersed gravity anomalies, while two basins (i.e. Forth Approaches Basin and Norwegian-Danish Basin) located in the south is in accordance with relatively smooth anomalies. The apparent positive-negative alternating patterns of D5 and D6 with the source depth of 13.7 km and 19.6 km are consistent with the crustal shearing and extrusion in the middle crust. D7 with the mean source depth of 27.0 km primarily reflects the Moho undulation. The D8 and A8 are smooth, corresponding to density distribution of the upper mantle. The detailed discussions are seen in Section 3.1 and 3.2 of the manuscript.

The motivation of this study is to develop a new parameterization of gravity field based SRBFs in the framework of MRR, and the wavelet decomposition and wavelet analysis are only used to separate the contributions of different anomaly sources, which are finally used to design the parameterizations of multiply layers. And, the detailed investigation of geological structure and geophysical mechanism in this area is out the scope of this study. The authors expect to make a detailed investigation of geological structure through the wavelet-based method based on gravity data over the North Sea, and compare it with the results derived from other data sources (e.g., GPS data and seismic wave data), and the raw manuscript is in preparation.

[5] Table 3. Please show the detailed algorithm and method to determine reasonably the depth of each layer.

Response: The authors thank the reviewer for this beneficial comment. The detailed algorithm and method to determine reasonably the depth of each layer can be referred to Spector & Grant (1970) and Xu et al. (2018). In the manuscript, we only show the primary equations (see Eq. (4) and Eq. (5)). The method is also presented in details as follows.

According to the solution to two-dimensional Lapalace's equation, each $D_w(\varphi, \lambda)$ of the eq. (4) in the manuscript can be expressed as (Spector and Grant, 1970; Syberg, 1972; Cianciara and Marcak, 1976):

$$D_w(\varphi, \lambda) = \sum_{\varphi} \sum_{\lambda} G_K e^{i2\pi(K_\varphi \varphi + K_\lambda \lambda)} e^{2\pi KH}$$

where $G_K$ denotes the amplitude, $K = \sqrt{K_\varphi^2 + K_\lambda^2}$ is the wave number, $(\varphi, \lambda)$ is the geodetic latitude and longitude, and $H$ is the elevation of $D_w(\varphi, \lambda)$. Thus, $G_K$ can be determined by:

$$G_K = \sum_{\varphi} \sum_{\lambda} D_w(\varphi,\lambda) e^{-i2\pi(K_\varphi \varphi + K_\lambda \lambda)} e^{\pm 2\pi KH}$$

When $H = 0$, the last equation can be written as:

$$(G_K)_0 = \sum_{\varphi} \sum_{\lambda} D_w(\varphi,\lambda) e^{-i2\pi(K_\varphi \varphi + K_\lambda \lambda)}$$

Inserting this equation into $G_K$

$$G_K = (G_K)_0 e^{\pm 2\pi KH}$$

Hence,

$$P_K = (P_K)_0 e^{\pm 4\pi KH}$$

where $P_K = (G_K)^2$ is the power. Then,

$$\ln P_K = \ln(P_K)_0 \pm 4\pi KH$$

in which $\ln P_K$ is natural logarithm of $P_K$. Based on the linear correlation between $K$ and $\ln P_K$, the corresponding average source depth $h_w$ of $D_w(\varphi,\lambda)$ can be estimated as (Spector and Grant, 1970; Xu et al., 2018)

$$h_w = \frac{1}{4\pi} \frac{\Delta \ln P_K^w}{\Delta K_w} \quad w = 1,2,\cdots,W$$

$\Delta \ln P_K^w$ and $\Delta K_w$ are the change rates for $\ln P_K^w$ and radial wave number $K_w$, respectively. In this manner, the corresponding average source depths $h_w$ of all decomposed wavelet details $D_w(\varphi,\lambda)$ $(w = 1,2,\cdots,W)$ and wavelet approximation $A_W(\varphi,\lambda)$ can be computed.

According to the reviewer's comment, we add the detailed algorithm and method to determine reasonably the depth of each layer, please see pp 7-8 in the revised manuscript.

[6] Section 3.3. We all know that the precision of terrestrial gravity data is better than that of ship-borne gravity data. But 1.45 mGal is for the terrestrial gravity data and

1.30 mGal for the ship-borne gravity data. Why?

Response: We thank the reviewer's for the beneficial comments. Yes, for the raw gravity observations, terrestrial data usually have better qualities than shipboard measurements. The estimated posterior variance factor of terrestrial data (1.45 mGal) is larger than that of shipboard data (1.30 mGal), we believe it is mainly due to the more significant uncorrected terrain effects in land than in ocean. We model the local gravity field based on the remove-compute-restore (RCR) method, and only the residual gravity field is parameterized through the SRBFs based on the single-layer or multilayer approach. Thus, the residual gravity data (after removing the GGM-derived components and RTM corrections) rather than the original data are used for modeling. Due to the limited spatial resolution of gravity measurements, we use the RTM to recover the local high-frequency signals that cannot be extracted from the gravimetry measurements. We see the local gravity field becomes smooth on land with RTM corrections, especially in areas with topography variation, however, uncorrected signals remain, e.g., see the red signals around Fraserburgh in England (around -2 °W and 57.5 °N) and blue ones along the boundary between France and Germany (around 8 °W and 49 °N) (see Figure 2 and Section 3.2 in Wu et al. (2017c)), which are mainly due to the limitations of the DTM (in terms of both spatial resolution and precision) and inaccuracy of the density parameter for the topography. These uncorrected high-frequency errors inevitably propagate into the regional solutions when modeling with single-layer/multilayer approach, see the data residuals showing in Figure 6 (Figure 5 in the original version); and the most significant residuals concentrate at these mountainous regions where the data are relatively sparse and uncorrected terrain corrections remains. On the other hand, the marine gravity field seems to be less affected by the local topographical effects, mainly due to the relatively small variation of local bathymetry (see Section 3.2 in Wu et al. (2017c)), and less significant residuals show in ocean parts, see Figure 6. The posterior variance factors are directly computed based on the data residuals, e.g., see eq.(25) in Klees et al. (2008), thus the estimated value for terrestrial observation group demonstrates lower accuracy.

[7] Figure 5. How to reduce effectively the edge effects for the local model?

Response: The authors thank the reviewer for the comment. The boundary limits for the area of Figure 5 are contracted by 0.5 ° in all the directions to reduce the edge effects. According to the reviewer's comment, we add this information (see pp 20) and redraw the data residuals only inside the boundary limits of this area, see Figure 6 in the updated version. Also, the corresponding statistics for data residuals are changed, see pp 20 and Table 5 in pp 25.

[8] Table 5. Why all means are 0 in the table?

Response: The authors thank the reviewer for this comment. We believe the mean values for the residuals should be zero, the local gravity field is modeled in the

framework of least squares system, both for the single-layer and multilayer approach. As a result, the sum of the residuals after least squares adjustment is zero, and the mean values for different observation groups derived from different approaches should be zero.

[9] Page 22. The degree for EGM2008 is up to 2190, but the order for EGM2008 is not up to 2190.

Response: The authors thank the reviewer for this beneficial comment. Yes, we believe the reviewer is right, the full order of EGM2008 is 2159 not 2190. According to the reviewer's comment, we correct this information, please see pp 26 in the updated version.

[10] Figures 7 and 8, Table 7. Why remove the mean differences?

Response: Thanks the reviewer for the comments. Due to the commission errors and uncorrected systematic errors in gravity data and inconsistencies among different height datums, the various GGMs (e.g., EGM2008 and EIGEN-6C4) deviate from local values observed from GPS/leveling data by tens of centimeters levels or even larger in the area of this study. While, the local gravity field model like QGNSea seems suffers less from this problem (see the mean values in Table 6), mainly due to the incorporation of more high-quality data and refined data preprocessing procedures. Thus, if we don't remove the mean differences between the GGMs and local GPS/leveling data, these differences are almost dominated by these systematic biases, which is undesirable for model comparisons. After removing these biases, the accuracies of different models can be clearly shown in Figure 9 (Figure 8 in the original version) and Table 7, and we also remove the mean values of the differences between QGNSea and other models to make these comparisons consistent, see Figure 8 (Figure 7 in the original version). According to the reviewer's comment, we add this information to the revised version, please see pp 27 in the updated version.

[11] Figure 8. We can see the systematical differences in the figure. Why?

Response: Thanks the reviewer for this comment. Yes, we agree with the reviewer's comment that the systematic errors remain. The regional gravimetric model usually deviates from the local GPS/leveling data, mainly due to commission errors in the GGMs and uncorrected systematic errors in the data and height systems. The systematic errors still exist in the results shown in Figure 9 (Figure 8 in the original version), since these errors cannot be thoroughly by simply removing the mean differences. As shown in pp 23-24, several methods can be used for reducing these systematic errors, and properly combine GPS/leveling data and gravimetric solution. However, the target for this study is to develop a multilayer approach for gravimetric

quasi-geoid modeling, which is served as a basic surface for further geophysical applications. While, after implementing these methods for combining local GPS/leveling and gravimetric model, the derived quasi-geoid is not purely gravimetric. Besides, the final solution may be distorted if only the locally distributed GPS/leveling data are combined, especially in the ocean parts, since no control data exist in these regions, see the detailed discussion in pp 23-24. Similar results for comparing the GPS/leveling data and gravimetric solutions can be found in Wu et al. (2017a, c). According to the reviewer's comment, we add this information in the revised manuscript, please see pp 27 in the updated version.

**A multilayer approach and its application to QGNSea V1.0 model: a local gravimetric quasi-geoid model over the North Sea: QGNSea V1.0**

**Yihao Wu**[1, 2], **Zhicai Luo** [3], **Bo Zhong**[4], **Chuang Xu**[2, 5]

[1] School of Earth Sciences and Engineering, Hohai University, Nanjing, China

[revised manuscript text omitted]